# Provable Watermarking for Data Poisoning Attacks

**Yifan Zhu**[1, 2]**, Lijia Yu**[3]*****, Xiao-Shan Gao**[1, 2]*****
[1]State Key Laboratory of Mathematical Sciences,
Academy of Mathematics and Systems Science, Chinese Academy of Sciences
[2]University of Chinese Academy of Sciences
[3]Institute of AI for Industries, Nanjing, China
zhuyifan@amss.ac.cn, ljyu@iaii.ac.cn, xgao@mmrc.iss.ac.cn

## Abstract

In recent years, data poisoning attacks have been increasingly designed to appear harmless and even beneficial, often with the intention of verifying dataset ownership or safeguarding private data from unauthorized use. However, these developments have the potential to cause misunderstandings and conflicts, as data poisoning has traditionally been regarded as a security threat to machine learning systems. To address this issue, it is imperative for harmless poisoning generators to claim ownership of their generated datasets, enabling users to identify potential poisoning to prevent misuse. In this paper, we propose the deployment of watermarking schemes as a solution to this challenge. We introduce two provable and practical watermarking approaches for data poisoning: *post-poisoning watermarking* and *poisoning-concurrent watermarking*. Our analyses demonstrate that when the watermarking length is $\Theta(\sqrt{d}/\epsilon_w)$ for post-poisoning watermarking, and falls within the range of $\Theta(1/\epsilon_w^2)$ to $O(\sqrt{d}/\epsilon_p)$ for poisoning-concurrent watermarking, the watermarked poisoning dataset provably ensures both watermarking detectability and poisoning utility, certifying the practicality of watermarking under data poisoning attacks. We validate our theoretical findings through experiments on several attacks, models, and datasets.

## 1 Introduction

Data poisoning [7, 43, 71] is a well-established security concern for modern ML systems. Its significance has become increasingly pronounced in the era of large-scale models, where many models are trained on web-crawl or synthetic data without rigorous selection [66, 10, 77]. There are two representative data poisoning attacks, *backdoor attacks* [13, 75] and *availability attacks* [37, 22]. Backdoor attacks involve creating poisoned datasets that cause models trained on them to predict the specific targets when a particular trigger is injected into test instances. Availability attacks aim to compromise model generalization by ensuring that models trained on poisoned datasets have low test accuracy. Deploying models on backdoor and availability attacked datasets poses severe security risks. For instance, in autonomous driving systems, triggered road signs created by backdoor attacks could be misclassified by object detectors, leading to potentially catastrophic accidents [27, 31]. Availability attacks directly undermine model utility, rendering AI-based systems nonfunctional [8].

However, interestingly, things are always two-faced. Modern data poisoning attacks are increasingly being designed to be harmless and purposeful. For example, backdoor attacks have been employed for black-box dataset ownership verification [50, 51], availability attacks have been utilized to prevent the unauthorized use of data [37, 23]. More recently, methods like NightShade [65] and Glaze [64] have been developed to protect artists' intellectual property from generative AI models. These

---

*Corresponding authors

advancements illustrate the promising potential of "data poisoning for good," transforming data poisoning attacks—traditionally viewed as harmful—into tools that can benefit society. Nevertheless, unintended consequences may arise. An innocent, authorized user might inadvertently use poisoned data, leading to potential misunderstandings and conflicts. To mitigate such risks, the poisoning generators must transparently disclose the presence of potential poisoning to their intended users. For example, when an artist distributes his works to a copyright protection system, the system (poisoner) not only aims to prevent unauthorized use but also bears the responsibility of informing clients and authorized users if the data has been perturbed. Such transparency is essential to ensure trust and avoid unintended harm in these beneficial applications of data poisoning.

To address the challenges and ensure the transparency of poisoned datasets, a direct approach is to design detection methods capable of identifying potential poisoning. While many studies have focused on detecting backdoor and availability attacks [11, 19, 18, 98, 89], these detection methods vary significantly across different types of attacks, making it challenging to unify as a single, cohesive framework for distribution to authorized users. Additionally, existing detection methods often rely on heuristic training algorithms, lacking a provable mechanism for claiming poisoning. This limitation can lead to disputes if a poisoned dataset is inadvertently misused, as the absence of a clear, verifiable claim undermines accountability. To overcome these challenges, we explore the use of watermarking [9, 1, 41], a widely adopted approach for copyright protection and the detection of AI-generated content, which presents a promising solution for poisoners to provably declare the existence of poisoning, thereby enhancing transparency and minimizing the risk of disputes.

In this paper, we propose two provable and practical watermarking approaches for data poisoning: *post-poisoning watermarking* and *poisoning-concurrent watermarking*. The former addresses scenarios where the poisoning generators require a third-party entity to create watermarks for their poisoned datasets, while the latter focuses on cases where the poisoning generators craft watermarks themselves. In Section 4.1, we demonstrate that when watermarking is sample-wise for each data, discernment of poisoned data with high probability is achievable if a specific key is available when the required watermarking length is $\Omega(\sqrt{d}/\epsilon_w)$ and $\Omega(1/\epsilon_w^2)$ for post-poisoning and poisoning-concurrent watermarking respectively ($d$ is the data dimension, $\epsilon_w$ is the watermarking budget). However, the sample-wise approach necessitates $N$ distinct watermarks and keys for a dataset with $N$ samples, which can be impractical for large datasets. To address this limitation, we consider a more meaningful scenario where a single watermark and key apply to all data instances. In Section 4.2, recognizing that reliance on the sample size $N$ is not ideal for universal watermarking, we extend our analysis to watermarking effective on most samples and then generalizes to the whole distribution with high probability. Specifically, we prove that when the post-poisoning and poisoning-concurrent watermarking lengths are $\Theta(\sqrt{d}/\epsilon_w)$ and $\Theta(1/\epsilon_w^2)$ respectively, the majority of poisoned data can be effectively identified. Moreover, if the sample size satisfies $N = \Omega(d)$, these results can be generalized to the entire data distribution.

Beyond demonstrating the effectiveness of watermarking, in Section 5, we further show that the injected watermarks have minimal impact on the poisoning. Specifically, for post-poisoning watermarking, when the data dimension $d$ and sample size $N$ are large, the generalization gap between the original poisoned distribution and the watermarked poisoned dataset is bounded by negligible terms. For poisoning-concurrent watermarking, achieving a small generalization gap requires an additional condition: the watermarking length should satisfy $O(\sqrt{d}/\epsilon_p)$, where $\epsilon_p$ is the poisoning budget.

Our theoretical analyses confirm that the effectiveness of watermarked data poisoning is maintained under specific watermarking lengths. For post-poisoning watermarking, both watermarking and poisoning remain effective when the length is $\Theta(\sqrt{d}/\epsilon_w)$. For the poisoning-concurrent watermarking, the effectiveness is certified when the length falls within the range of $\Theta(1/\epsilon_w^2)$ to $O(\sqrt{d}/\epsilon_p)$. Consequently, if the poisoning generator relies on a third-party entity for watermarking, using a larger length is advantageous. In comparison, if the generator directly embeds the watermark into their poisoned dataset, a moderate length is more practical. In Section 6, we evaluate several existing backdoor and availability attacks to empirically validate our theoretical findings.

## 2 Related Work

**Data Poisoning.** Data poisoning attacks modify the training data within a small perturbation budget, aiming to elicit unusual behaviors for models trained on the poisoned dataset. One prominent type of

data poisoning is backdoor attacks [13, 27, 79, 80, 95, 52, 62, 75, 55, 91, 88]. These attacks inject specific patterns into the training data, causing the trained model to behave anomalously when test instances contain such patterns. Other works [50, 51] have utilized backdoor attacks to achieve dataset ownership verification. Another category is availability attack, also referred to as indiscriminate attacks [7, 59, 21, 22, 44, 53, 86]. These attacks aim to degrade the model's overall test accuracy. Recently, unlearnable examples [37, 23, 32, 70, 12, 67, 92, 97, 83] as the case of imperceptible availability attacks, have been designed to protect data from illegal use by unauthorized trainers. Further data poisoning schemes include targeted attacks [43, 71, 30, 25, 4], which cause models to malfunction on some specific data. In this paper, we mainly focus on imperceptible clean-label backdoor attacks and availability attacks, as they are more practical in real-world scenarios.

**Watermarking.** Watermarking involves embedding special signals into training data or models to enhance copyright protection and identify data ownership [61, 5, 40, 96, 3, 73, 82]. People [50, 51] introduced backdoor attacks as the dataset watermark for data verification, while [29] proposed the domain watermark with harmless verification. Recently, watermarking of large language models has gained significant attention for AI-generated text detection [41, 35, 47, 42, 93, 15]. Watermarking has also been investigated for generative image models [85, 93, 28]. This paper focuses on watermarking for poisoning attacks. We provide two provable, simple, and practical watermarking schemes: post-poisoning and poisoning-concurrent watermarking. To the best of our knowledge, this is the first work to leverage watermarking schemes in the context of data poisoning attacks.

# 3 Preliminaries

## 3.1 Data Poisoning

We assume the data always lies in $[0,1]^d$. To ensure consistency across different criteria, we focus on imperceptible clean-label data poisoning attacks, which are more practical in real-world applications. Specifically, we denote the attack as a mapping $\delta^p : [0,1]^d \to [-\epsilon_p, \epsilon_p]^d$. For each data $x$, the attack $\delta^p$ perturbs the data to produce a modified version $x' = x + \delta^p(x)$, while ensuring $\|\delta^p(x)\|_\infty \le \epsilon_p$ to preserve imperceptibility. For simplicity, we denote $\delta^p(x)$ as $\delta_x^p$.

**Goal.** The poisoning objective risk is defined as $\mathcal{R}^{\text{poi}}(\mathcal{D}^{\text{victim}}, \mathcal{F})$, where $\mathcal{D}^{\text{victim}}$ represents the victim distribution. The goal of data poisoning is to construct a poisoned distribution $\mathcal{D}'$, such that if the risk $\mathcal{R}(\mathcal{D}', \mathcal{F}) = \mathbb{E}_{(x,y)\sim\mathcal{D}'}\mathcal{L}(\mathcal{F}(x), y)$ is small, the network $\mathcal{F}$ achieves a small poisoning objective risk $\mathcal{R}^{\text{poi}}(\mathcal{D}^{\text{victim}}, \mathcal{F})$. In other words, when $\mathcal{F}$ has effectively learned the poison features of $\mathcal{D}'$, it is expected to exhibit specific properties aligned with the objectives of data poisoning. For example, in availability attacks, $\mathcal{D}^{\text{victim}} = \mathcal{D}$, the objective risk $\mathcal{R}^{\text{poi}}(\mathcal{D}^{\text{victim}}, \mathcal{F}_{S'}) = \mathbb{E}_{(x,y)\sim\mathcal{D}}[-\mathcal{L}(\mathcal{F}(x), y)]$, where the goal is to obtain a network $\mathcal{F}$ with high loss on $\mathcal{D}$, thereby degrading its generalization performance. In backdoor attacks, $\mathcal{D}^{\text{victim}} = \mathcal{D} \oplus T$, $\mathcal{R}^{\text{poi}}(\mathcal{D}^{\text{victim}}, \mathcal{F}_{S'}) = \mathbb{E}_{(x,y)\sim\mathcal{D}}[\mathcal{L}(\mathcal{F}(x \oplus T), y^{\text{target}})]$, where $T$ is the trigger injected during inference, $y^{\text{target}}$ is the targeted label. The goal of backdoor attacks is to ensure that any data $x$ with trigger $T$ be classified as $y^{\text{target}}$.

## 3.2 Watermarking

**Watermark and key.** The goal of watermarking on data poisoning attacks is to ensure that verified users are aware of whether the given data has been poisoned, to prevent potential misunderstandings when data creators use data poisoning attacks to achieve specific objectives. e.g., crafting unlearnable examples to deter unauthorized use of data. In this paper, we mainly focus on dataset watermarking [51], where watermarks are embedded in datasets for verification. Specifically, similar to the data poisoning attack $\delta^p$, we denote a watermarking as a mapping $\delta^w : [0,1]^d \to [-\epsilon_w, \epsilon_w]^q$, and use $\delta_x^w$ to represent $\delta^w(x)$ for simplicity, where $q \le d$ is the watermarking length, and the watermarking dimension indices are $\mathcal{W} = \{d_1, d_2, \cdots, d_q\} \subset [d]$. When a dataset is watermarked, authorized users are provided with a corresponding key to detect whether the data contains watermarks. In this paper, we assume that the key $\zeta$ is a $d$-dimensional vector. The watermarking detector uses a simple mechanism, computing the inner product $\zeta^T x$ to determine whether $x$ has been watermarked.

**Post-poisoning watermarking.** In this scenario, a third-party entity serves as the watermark generator, crafting watermarks for a given poisoned dataset. The goal is to enable authorized detectors to identify potential poisoned data. Denote the poison and the watermark as $\delta_x^p$ and $\delta_x^w$ respectively,

where $\|\delta_x^w\|_\infty \leq \epsilon_w$, $\|\delta_x^p\|_\infty \leq \epsilon_p$. Both watermark $\delta_x^w$ and poison $\delta_x^p$ rely on data $x$, and the overall perturbation is $\delta_x = \delta_x^p + \delta_x^w$. For simplicity, we denote the perturbation for data $x_i$ as $\delta_i = \delta_{x_i}$.

**Poisoning-concurrent watermarking.** In this scenario, the watermark generator also acts as the poison generator, simultaneously crafting both watermarks and poisons. The objective is to achieve the goals of data poisoning while ensuring authorized detectors can identify the poisoned data. Since the watermark generator can control the poison dimensions, we assume the generator separates the dimensions used for watermarking and poisoning. Specifically, the dimensions for poisoning are indexed by $\mathcal{P} = [d] \backslash \mathcal{W}$. Other notations remain consistent with those at post-poisoning watermarking.

To make notations clearer, we provide a symbol table in Appendix A.

### 3.3 A Practical Threat Model

Any copyright owner can deploy our watermarking when releasing their original datasets to a third party (e.g., AI training platforms, academic institutions, and copyright certification systems). To make the threat model more concrete, we provide a detailed deployment scenario below.

A company (called Alice) that collects a large proprietary dataset for autonomous driving research (e.g., dash cam video frames). She wants to open source a part of her dataset to promote innovation for the community (e.g., Non-profit research organization), but also wants to prevent unlicensed users from training a machine learning model on it successfully to protect her intellectual property. To achieve the above goals, Alice runs our poisoning + watermarking algorithm on every instance of her dataset, publishing the perturbed (i.e., protected) dataset which is unlearnable by standard models and obtains a secret, key-dependent watermark signal. She publishes this on her GitHub under a permissive license, accompanied by a SHA256 hash so any recipient can verify integrity.

A research lab (called Bob) registers on Alice's portal and agrees to a standard agreement for legal use of the dataset. After approval by Alice, Bob receives a secret key (e.g., a 128 bit seed) provided via Alice's portal's secure HTTPS channel. Furthermore, Bob also gains a pipeline (e.g., Python pre-processing package) from Alice such that he can run the watermark detection to verify his identity and ensure that there is no file corruption. After the verification, Bob can run an algorithm designed by Alice (e.g., directly adding inverse unlearnable noise for each data) to remove the unlearnable poisons. If the pipeline receives the wrong key or a tampered file, the detection fails and the poisons cannot be removed to ensure the unlearnability.

For a malicious user (called Chad), first, Chad can download the same public poisoned and watermarked dataset, but cannot train a good model on it because the dataset is unlearnable. If Chad tries to remove or tamper with watermarks and unlearnable poisons without knowing the secret key, detection will fail.

For key management, Alice can rotate keys per month and publish on her portal only to approved accounts (i.e., trusted users). Alice can also add a HMAC scheme to prevent potential forgery risks. Specifically, Alice can rotate keys per month and publish on her portal only to approved accounts (i.e., trusted users). Alice can also add a HMAC scheme to prevent potential forgery risks. Specifically, we can separate keys into generation key $k_{gen}$ and authentication key $k_{auth}$, where $k_{gen}$ is completely the same as our paper and correlates with injected watermarks $w_i$ for every data $x_i$. For each perturbed $\hat{x}_i$ with $w_i$, we can compute an additional tag $t_i$ by HMAC under $k_{auth}$, i.e., $t_i = \mathrm{HMAC}_{k_{auth}}(id_i, \hat{x}_i)$, where $id_i$ is a unique identifier for the image $x_i$ (e.g., index). After that, we store the $(id_i, t_i)$ pair (e.g., through a sidecar JSON) for later detection. In watermarking detection, beyond traditional detection using $k_{gen}$ and $\hat{x}_i$, we also verify the tag with $t_i$ and $t_i = \mathrm{HMAC}_{k_{auth}}(id_i, \hat{x}_i)$ to avoid potential forgery attacks. In this case, even if the generation key $k_{gen}$ leaks, an attacker cannot forge a new valid $(x_i, t_i)$ pair as they lack the authentication key $k_{auth}$. We can keep $k_{auth}$ in a secure enclave and rotate it independently with $k_{gen}$ to enhance the security.

## 4 Soundness of Watermarking

In this section, we provide theoretical guarantees for the conditions under which watermarking can effectively differentiate between poisoned and benign data. We begin by examining a specific version where the watermarking is sample-wise. In this case, the injected watermark $\delta_x^w$ relies on $x$, meaning that the watermark generator can assign a unique watermark to each data.

## 4.1 Sample-wise Version

We first analyze the sample-wise version of post-poisoning watermarking. Proofs of theorems in this subsection are provided in Appendix B.1.

**Theorem 4.1** (Sample-wise, post-poisoning watermarking). *For any data point $x$ sampled from $\mathcal{D}_\mathcal{X}$ and their corresponding poison be $\delta_x^p$, there exists a distribution $\Xi$ defined in $\mathbb{R}^d$ such that we can sample the key $\zeta \sim \Xi$ satisfying that for any $\omega \in (0,1)$, there are:*

*(1):* $\mathbb{P}_{x \sim \mathcal{D}_\mathcal{X}, \zeta \sim \Xi} \left( \zeta^T x < \sqrt{\frac{d}{2} \log \frac{1}{\omega}} \right) > 1 - w$; *(2): we can craft the watermark $\delta_x^w$ based on $\zeta$ such that* $\mathbb{P}_{x \sim \mathcal{D}_\mathcal{X}, \zeta \sim \Xi} \left( \zeta^T (x + \delta_x) > q\epsilon_w - \sqrt{\frac{d}{2} \log \frac{1}{\omega}} \right) > 1 - w$. *Hence, when $q > \frac{1}{\epsilon_w} \sqrt{2d \log \frac{1}{\omega}}$, it holds that* $\mathbb{P}_{x_1, x_2 \sim \mathcal{D}_\mathcal{X}, \zeta \sim \Xi} \left( \zeta^T (x_1 + \delta_1) > \zeta^T x_2 \right) > 1 - 2\omega$.

*Remark* 4.2. For the sample-wise, post-poisoning watermarking with the data length $d$ and watermark budget $\epsilon_w$, crafting an effective watermark requires the watermarking length to be $\Omega(\sqrt{d}/\epsilon_w)$.

Next, we analyze the scenario of sample-wise version for poisoning-concurrent watermarking.

**Theorem 4.3** (Sample-wise, poisoning-concurrent watermarking). *For any $x \sim \mathcal{D}_\mathcal{X}$, there exists a distribution $\Xi \in \mathbb{R}^d$ such that we can sample the key $\zeta \sim \Xi$ satisfied that for any $\omega \in (0,1)$:*

*(1):* $\mathbb{P}_{x \sim \mathcal{D}_\mathcal{X}, \zeta \sim \Xi} \left( \zeta^T x < \sqrt{\frac{q}{2} \log \frac{1}{\omega}} \right) > 1 - w$; *(2): we can craft the watermark $\delta_x^w$ and poison $\delta_x^p$ such that* $\mathbb{P}_{x \sim \mathcal{D}_\mathcal{X}, \zeta \sim \Xi} \left( \zeta^T (x + \delta_x) > q\epsilon_w - \sqrt{\frac{q}{2} \log \frac{1}{\omega}} \right) > 1 - w$. *Hence, when $q > \frac{2}{\epsilon_w^2} \log \frac{1}{\omega}$, it holds that* $\mathbb{P}_{x_1, x_2 \sim \mathcal{D}_\mathcal{X}, \zeta \sim \Xi} (\zeta^T (x_1 + \delta_1) > \zeta^T x_2) > 1 - 2\omega$.

*Remark* 4.4. For the sample-wise, poisoning-concurrent watermarking with the data length $d$ and watermark budget $\epsilon_w$, crafting an effective watermark requires the watermarking length to be $\Omega(1/\epsilon_w^2)$.

*Remark* 4.5. The required length for poisoning-concurrent watermarking $\Omega(1/\epsilon_w^2)$ is smaller than that for post-poisoning watermarking $\Omega(\sqrt{d}/\epsilon_w)$. This difference arises because the condition $q \leq d$ for the watermarking length always holds. Therefore, we have $\epsilon_w \geq O(1/\sqrt{d})$, which implies $\Omega(\sqrt{d}/\epsilon_w) \geq \Omega(1/\epsilon_w^2)$.

Theorems 4.1 and 4.3 suggest that, with high probability, as long as the watermark dimension $q$ reaches the required thresholds ($\Omega(\sqrt{d}/\epsilon_w)$ or $\Omega(1/\epsilon_w^2)$), the inner product of key and poisoned data will exceed a constant $C_1$, while the inner product of key and clean data will remain below a constant $C_2 < C_1$. As a result, a detector can simply select a threshold $T = \frac{C_1 - C_2}{2}$ to effectively differentiate between poisoned and clean data using the given key. Based on these observations, we derive the following corollary:

**Corollary 4.6.** *For sample-wise, post-poisoning watermarking, if the watermarking length $q \geq \frac{2}{\epsilon_w} \sqrt{2d \log \frac{1}{\omega}}$, with probability at least $1 - 2\omega$, for the sampled key $\zeta \in \mathbb{R}^d$ and data $x_1, x_2$ sampled from $\mathcal{D}_\mathcal{X}$, it is possible to craft the watermark $\delta_x^w$ such that $\zeta^T (x_1 + \delta_1) > \frac{3}{4} q\epsilon_w, \zeta^T x_2 < \frac{1}{4} q\epsilon_w$. Similarly, for poisoning-concurrent watermarking, if $q \geq \frac{8}{\epsilon_w^2} \log \frac{1}{\omega}$, we can craft the watermark $\delta_x^w$ such that $\zeta^T (x_1 + \delta_1) > \frac{3}{4} q\epsilon_w, \zeta^T x_2 < \frac{1}{4} q\epsilon_w$.*

However, the sample-wise watermarking requires an individual key for each sample, which is impractical in real-world applications. A more ideal case is that the watermark detector can use a single key applicable to all samples, making detection more effective and efficient. This motivates the consideration of the universal version of watermarking, where the injected watermark $\delta_x^w$ is identical for every $x$. In this case, scenario, for simplicity, we denote $\delta_x^w = \delta^w$.

## 4.2 Universal Version

In the universal version, a single detection key is employed, violating the condition of Theorems 4.1 and 4.3, where the key $\zeta$ is sampled from a distribution. Consequently, the proof techniques used for the sample-wise case are difficult to generalize to this scenario. Instead, we step away from the distributional guarantees and first analyze the finite-sample case. The theoretical results for the finite case can subsequently be extended to the distributional setting. Proof of theorems in this subsection

are in Appendix B.2. In the finite case, we assume the dataset consists of $N$ samples, denoted as $S_\mathcal{X} = \{x_1, x_2, \cdots, x_N\}$. We begin by analyzing the case of post-poisoning watermarking.

**Proposition 4.7** (Universal, post-poisoning watermarking)**.** *For the dataset $S_\mathcal{X}$, when $q > \frac{2+\epsilon_p}{\epsilon_w}\sqrt{\frac{d}{2}\log\frac{2N}{\omega}}$, we can sample the key $\zeta \in \mathbb{R}^d$ from a certain distribution such that, with probability at least $1 - \omega$, there exists the watermark $\delta^w$ such that $\zeta^T(x_j + \delta_j) > \zeta^T x_i, \forall i, j \in [N]$.*

We can extend the proof of Proposition 4.7 with a larger $q$, to achieve a non-vacuous gap between poisoned and benign data, as stated in the following corollary:

**Corollary 4.8.** *Notations are similar to Proposition 4.7. When $q > \frac{4}{\epsilon_w}\sqrt{\frac{d}{2}\log\frac{2N}{\omega}}$ with probability at least $1 - \omega$, there exists the watermark $\delta^w$ such that $\zeta^T(x_i + \delta_i) > \frac{q\epsilon_w}{2}, \zeta^T x_i < \frac{q\epsilon_w}{4}, \forall i \in [N]$.*

Proposition 4.7 demonstrates that the watermarking length is expected to be $\Omega(\sqrt{d\log N}/\epsilon_w)$ to ensure universal watermarking discerning every data $x_i$, which is not ideal as the watermarking length $q$ depends on sample size $N$, leading to vacuous results when the dataset becomes sufficiently large. To address this limitation and achieve a non-vacuous result, we propose relaxing the properties from discerning every sample to discerning most samples, as described in the following theorem.

**Theorem 4.9** (Universal, post-poisoning watermarking for most examples)**.** *For the dataset $S_\mathcal{X} = \{x_1, x_2, \cdots, x_N\}$, $x_i$ and the poison $\delta_p^i$ are i.i.d. sampled from $\mathcal{D}_\mathcal{X}$ and $\mathcal{D}_\mathcal{P}$ respectively. For any $w \in (0, 1/2)$ and $q > \frac{2}{\epsilon_w}\sqrt{2d\log\frac{1}{\omega}}$, we can sample the key $\zeta \in \mathbb{R}^d$ from a certain distribution such that, with probability at least $1 - 2\exp\left(\frac{-N(\omega - e^{-q^2\epsilon_w^2/8d})^2}{\omega + e^{-q^2\epsilon_w^2/8d}}\right)$, we can craft the watermark $\delta^w$, such that $\zeta^T(x_i + \delta_i) > \frac{q\epsilon_w}{2}, \zeta^T x_i < \frac{q\epsilon_w}{4}$ holds for at least $(1 - 2\omega)N$ samples.*

*Remark* 4.10. Theorem 4.9 suggests that when the sample size $N$ is sufficiently large and the watermark length $q \gtrsim \frac{2}{\epsilon_w}\sqrt{2d\log\frac{1}{\omega}} = \Theta(\sqrt{d}/\epsilon_w)$, the universal watermarking is effective for most samples with high probability. Thus, if we relax the requirement and only demand that the watermarking is effective for most samples, Theorem 4.9 indicates that the required watermarking length no longer depends on $N$, unlike in Proposition 4.7.

We then analyze the finite universal case for poisoning-concurrent watermarking.

**Proposition 4.11** (Universal, poisoning-concurrent watermarking)**.** *For the dataset $S_\mathcal{X}$, when $q > \frac{4}{\epsilon_w^2}\log\frac{N}{\omega}$, it is possible to sample the key $\zeta \in \mathbb{R}^d$ from a certain distribution such that, with probability at least $1 - \omega$, we can craft watermark $\delta^w$ and poison $\delta^p$ such that $\zeta^T(x_j + \delta_j) > \zeta^T x_i, \forall i, j \in [N]$.*

Similar to post-poisoning watermarking, we can derive analogous results for poisoning-concurrent watermarking about non-vacuous gaps and cases on most examples.

**Corollary 4.12.** *Notations are similar to Prop 4.11. When $q > \frac{9}{\epsilon_w^2}\log\frac{N}{\omega}$, with probability at least $1 - \omega$, we can craft watermark $\delta^w$ and poison $\delta^p$, such that $\zeta^T(x_i + \delta_i) > \frac{2q\epsilon_w}{3}, \zeta^T x_i < \frac{q\epsilon_w}{3}, \forall i \in [N]$.*

**Theorem 4.13** (Universal, poisoning-concurrent watermarking for most examples)**.** *For the dataset $S_\mathcal{X} = \{x_1, x_2, \cdots, x_N\}$, where $x_i$ is i.i.d. sampled from $\mathcal{D}_\mathcal{X}$. For any $\omega \in (0, 1)$ and $q > \frac{9}{2\epsilon_w^2}\log\frac{1}{\omega}$, we can sample the key $\zeta \in \mathbb{R}^d$ from a certain distribution such that, with probability at least $1 - \exp\left(\frac{-N(\omega - e^{-2q\epsilon^2/9})^2}{\omega + e^{-2q\epsilon^2/9}}\right)$, we can craft the watermark and the poison satisfies $\zeta^T(x_i + \delta_i) > \frac{2q\epsilon_w}{3}, \zeta^T x_i < \frac{q\epsilon_w}{3}$ holds for at least $(1 - \omega)N$ samples.*

*Remark* 4.14. Theorem 4.13 indicates that for a sufficiently large $N$ and $q \gtrsim \frac{9}{2\epsilon_w^2}\log\frac{1}{\omega} = \Theta(1/\epsilon_w^2)$, the universal, poisoning-concurrent watermarking is effective for most samples with high probability. Compared with Proposition 4.11, the condition of the watermarking length $q$ in Theorem 4.13 is independent of the sample size $N$.

After establishing results for the finite case for most samples, we can extend these guarantees to the entire data distribution, as presented in the following theorem.

**Theorem 4.15** (Generalization of universal watermarking to distributional case)**.** *For the dataset $S_\mathcal{X} = \{x_1, x_2, \cdots, x_N\}$, data $x_i$ and poison $\delta_p^i$ are i.i.d. sampled from $\mathcal{D}_\mathcal{X}$ and $\mathcal{D}_\mathcal{P}$ respectively.*

*Consider a universal watermark $\delta^w$, with probability at least $1 - 2\mu$ for the sampled data and poisons, if there exists a key $\zeta$ that satisfies $\zeta^T(x_i + \delta_i) > C_1, \zeta^T x_i < C_2$, for at least $(1 - \omega)N$ samples $x_i$, it has*

$$\mathbb{P}_{x,\tilde{x} \sim \mathcal{D}_{\mathcal{X}}, \delta^p \sim \mathcal{D}_{\mathcal{P}}}\left(\left\{\zeta^T(x + \delta^p + \delta^w) > C_1, -\zeta^T \tilde{x} < C_2\right\}\right) > 1 - 2\omega - 2\sqrt{\frac{d}{N}\left(\log\frac{2N}{d} + 1\right) - \frac{1}{N}\log\frac{\mu}{4}}.$$

*Remark* 4.16. When the sample size $N$ is greater than $\Omega(d)$, the effectiveness of watermarks in the finite case can, with high probability, be generalized to the distributional case.

Generalizing the universal watermarking from finite cases to distributional cases does not impose additional conditions on the watermarking length $q$. For universal, post-poisoning watermarking, as noted in Remark 4.10, an effective watermark for the distribution $\mathcal{D}_{\mathcal{X}}$ exists when $q = \Theta(\sqrt{d}/\epsilon_w), N = \Omega(d)$. For universal, poisoning-concurrent watermarking, as noted in Remark 4.14, an effective watermark exists when $q = \Theta(1/\epsilon_w^2), N = \Omega(d)$. Compared with sample-wise watermarking, achieving effective universal watermarking for a data distribution does not require more watermarking length $q$. The only additional requirement is that the dataset size $N$ is not too small (at least $\Omega(d)$), which is a reasonable condition for generalization in practical scenarios.

## 5 Soundness of Poisoning under Watermarking

In this section, we prove that poisoning remains effective under watermarking for an $L$-layer feed-forward neural network. For simplicity, we focus on universal watermarking as it is more practical; similar properties also apply to sample-wise watermarking. To facilitate theoretical analyses, we adopt the widely used Xavier normalization [26] for network parameters, which is also employed in Neural Tangent Kernel (NTK) [39] and many other theoretical works [20, 38, 87, 74, 63]. The proofs for this section are provided in Appendix B.3.

Assume the (normalized) $L$-layer feed-forward neural network is $\mathcal{F} : \mathbb{R}^d \to \mathbb{R}$ defined as $\mathcal{F}(x) = W^L \frac{1}{\sqrt{d_{L-1}}}\text{ReLU}(W^{L-1} \cdots \frac{1}{\sqrt{d_2}}\text{ReLU}(W^2 \frac{1}{\sqrt{d_1}}\text{ReLU}(W^1 x + b^1) + b^2) + \cdots + b^{L-1}) + b^L$ where $\text{ReLU}(x) = \max(0, x)$ is the activation function, $W^l \in \mathbb{R}^{d_l \times d_{l-1}}$ and $b^l \in \mathbb{R}^{d_l}$ are the weight matrix and the bias term of the $l$-th layer respectively for $l \in [L]$. We consider a binary classification task where the data distribution $\mathcal{D} \in [0,1]^d \times \{-1, 1\}$. Here $d_0 = d$ and $d_L = 1$. We also assume that $d_1 \geq d$ as modern neural networks are typically larger and tend to be overparameterized [46, 6, 10, 2]. The loss function used is the cross-entropy loss: $\mathcal{L}(\mathcal{F}(x), y) = \log(1 + e^{-y \cdot \mathcal{F}(x)})$.

**Definition 5.1** (Optimal Classifier). We define the optimal classifier for a dataset $S$ under the hypothesis space $\mathcal{F}$ as $\mathcal{F}_S^* = \arg\min_{\mathcal{F}} \frac{1}{|S|}\sum_{(x,y) \in S} \mathcal{L}(\mathcal{F}(x), y)$, where $\mathcal{L}$ is the loss function.

**Theorem 5.2** (Impact of Watermarking). *With probability at least $1 - 2\omega$ for the poisoned dataset $\{(x_i', y_i)\}_{i=1}^N = S' \sim \mathcal{D}'$ and the key $\zeta \in \mathbb{R}^d$ selected from a certain distribution, we can craft the watermark $\delta^w$ satisfying:*

$$\mathcal{R}(\mathcal{D}', \mathcal{F}_{S'+\delta^w}^*) \leq \mathbb{E}_\eta \frac{1}{N}\sum_{i=1}^N \mathcal{L}(\mathcal{F}_{S'}^*(x_i' + \eta), y_i)$$
$$+ O\left(\sqrt{\frac{L}{N}}\right) + O\left(\sqrt{\frac{\log d}{N}}\right) + O\left(\sqrt{\frac{\log 1/\omega}{N}}\right) + O\left(\epsilon_w \sqrt{\frac{q \log 1/\omega}{d}}\right),$$

*where $S' + \delta^w = \{(x_i' + \delta^w, y_i)\}_{i=1}^N$ is the watermarked dataset, $\eta \sim \mathcal{U}\{-\epsilon_w, \epsilon_w\}^q$ is a random vector.*

*Remark* 5.3. Since $\eta$ is a random noise under budget $\epsilon_w$, the optimal classifier $\mathcal{F}_{S'}^*$ tends to have small loss under perturbation $\eta$, resulting in $\mathbb{E}_\eta L(\mathcal{F}_{S'}^*(x_i + \eta), y_i)$ being small. Furthermore, if $d$ and $N$ are large enough, four error terms in Theorem 5.2 are all small when the post-poisoning condition in Section 4.2, $q = \Theta(\sqrt{d}/\epsilon_w)$ holds, resulting in a small $\mathcal{R}(\mathcal{D}', \mathcal{F}_{S'+\delta^w}^*)$.

To ensure the soundness of watermarked poisoning, we assume that the (un-watermarked) poisoning distribution $\mathcal{D}'$ is effective. First, we provide the definition of an effective poisoning distribution.

**Assumption 5.4** (($\lambda, \mu$)-effective poisoning distribution). A poisoning distribution $\mathcal{D}'$ is called ($\lambda, \mu$)-effective (for victim distribution $\mathcal{D}^{\text{victim}}$ and poisoning objective risk $\mathcal{R}^{\text{poi}}$), if $\mathcal{R}^{\text{poi}}(\mathcal{D}^{\text{victim}}, \mathcal{F}) \leq \lambda$ holds for network $\mathcal{F}$ where $\mathcal{R}(\mathcal{D}', \mathcal{F}) \leq \mu$.

Table 1: The clean accuracy (Acc,%), attack success rate (ASR,%), and AUROC of Narcissus and AdvSc backdoor attacks on both post-poisoning watermarking and poisoning-concurrent watermarking with different watermarking length $q$ under ResNet-18 and CIFAR-10.

| Length/Method | Narcissus [91] | | AdvSc [88] | |
| Acc/ASR/AUROC(↑) | Post-Poisoning | Poisoning-Concurrent | Post-Poisoning | Poisoning-Concurrent |
| --- | --- | --- | --- | --- |
| 0(Baseline) | 94.69/95.04/- | 94.69/95.04/- | 92.80/95.53/- | 92.80/95.53/- |
| 100 | 94.55/93.01/0.5522 | 95.12/91.30/0.9294 | 93.34/98.23/0.8036 | 92.91/96.81/0.9679 |
| 300 | 94.38/91.34/0.8226 | 94.61/96.47/0.9778 | 92.82/96.48/0.8779 | 93.05/95.23/0.9955 |
| 500 | 94.95/93.11/0.9509 | 94.70/95.03/0.9968 | 93.18/97.43/0.9218 | 92.89/95.79/0.9986 |
| 1000 | 94.40/92.43/0.9974 | 94.32/92.03/0.9992 | 93.05/94.41/0.9809 | 93.38/84.39/0.9995 |
| 1500 | 93.90/91.05/0.9997 | 94.67/80.60/1.0000 | 93.46/90.85/0.9959 | 93.11/56.11/1.0000 |
| 2000 | 94.55/90.37/1.0000 | 94.89/22.46/1.0000 | 93.40/79.97/0.9994 | 92.38/30.05/1.0000 |
| 2500 | 94.81/93.30/1.0000 | 94.67/11.86/1.0000 | 92.78/82.89/1.0000 | 92.65/12.14/1.0000 |
| 3000 | 94.93/90.02/1.0000 | 94.72/ 9.75/1.0000 | 93.10/74.82/1.0000 | 93.04/ 9.97/1.0000 |

To quantify the performance of the poisoning algorithm, we measure how well the network $\mathcal{F}$ has learned poison features and achieves the poisoning objective by $\mathcal{R}(\mathcal{D}', \mathcal{F}) \leq \mu$ and $\mathcal{R}^{\mathrm{poi}}(\mathcal{D}^{\mathrm{victim}}, \mathcal{F}) \leq \lambda$ respectively. In practice, an effective poisoning method should generate $(\lambda, \mu)$-effective poisoning distribution with small $\mu$ and $\lambda$. It is reasonable to assume that $(\lambda, \mu)$-effective poisoning distribution $\mathcal{D}'$ can be generated by some existing heuristic algorithm. For example, previous works [69, 98] have demonstrated that victim models with low test accuracy (small $\lambda$) learn poisoning features well (small $\mu$) under availability attacks. By Theorem 5.2, if $N$ and $d$ are large, $\mathcal{R}(\mathcal{D}', \mathcal{F}^*_{S'+\delta^w})$ is small enough, thus a well-trained network on the watermarked dataset $S' + \delta^w$ will result in lower $\mathcal{R}(\mathcal{D}', \mathcal{F})$, ensuring the soundness of post-poisoning watermarking through the following corollary:

**Corollary 5.5** (Post-poisoning watermarking). *If $\mathcal{D}'$ is a $(\lambda, \mu)$-effective poisoning distribution for some $\mu > 0$, when $N$ and $d$ are sufficiently large, with high probability, network $\mathcal{F}$ trained on post-poisoning watermarking dataset $S' + \delta_w = \{(x_i' + \delta^w, y_i)\}_{i=1}^N$ holds that $\mathcal{R}^{\mathrm{poi}}(\mathcal{D}^{\mathrm{victim}}, \mathcal{F}) \leq \lambda$.*

However, when we consider poisoning-concurrent watermarking, the dimension of the poisons $\delta^p$ is restricted under $\mathcal{P} \subset [d]$. In this case, we need to further bound the risk of $\mathcal{D}'$ under the restricted poisoned dataset $S'|_{\mathcal{P}} = \{(x_i + \delta_i^p|_{\mathcal{P}}, y_i)\}_{i=1}^N$, which induces the following theorem:

**Theorem 5.6** (Impact of Poisoning dimension). *With probability at least $1 - \omega$ of the (unrestricted) poisoned dataset $\{(x_i + \delta_i^p, y_i)\}_{i=1}^N = S' \sim \mathcal{D}'$, it holds that*

$$\mathcal{R}(\mathcal{D}', \mathcal{F}^*_{S'|_{\mathcal{P}}}) \leq \frac{1}{N}\sum_{i=1}^N \mathcal{L}(\mathcal{F}^*_{S'|_{\mathcal{P}}}(x_i + \delta_i^p|_{\mathcal{P}}), y_i) + O\left(\frac{q\epsilon_p}{\sqrt{d}}\right) + O\left(\sqrt{\frac{\log d}{N}}\right) + O\left(\sqrt{\frac{L}{N}}\right) + O\left(\sqrt{\frac{\log 1/\omega}{N}}\right).$$

In the case of poisoning-concurrent watermarking, if $N$ is large, and $q = O\left(\sqrt{d}/\epsilon_p\right)$, then $\mathcal{R}(\mathcal{D}', \mathcal{F}^*_{S'|_{\mathcal{P}}})$ becomes sufficiently small. Thus, a well-trained network $\mathcal{F}$ on a restricted poisoned dataset $S'|_{\mathcal{P}}$ tends to have a small risk under the (unrestricted) poisoning distribution $\mathcal{D}'$. Therefore, combined with Theorem 5.2, we can directly obtain the following corollary:

**Corollary 5.7.** *With probability at least $1 - 3\omega$ for the restricted poisoned dataset $S'|_{\mathcal{P}} \sim \mathcal{D}'|_{\mathcal{P}}$ and the key $\zeta \in \mathbb{R}^d$ selected from a certain distribution, we can craft the watermark $\delta^w$ satisfying:*

$$\mathcal{R}(\mathcal{D}', \mathcal{F}^*_{\tilde{S}}) \leq \mathbb{E}_\eta \frac{1}{N} \sum_{i=1}^N \mathcal{L}\left(\mathcal{F}^*_{S|_{\mathcal{P}}}(x_i + \delta_i^p|_{\mathcal{P}} + \eta), y_i\right)$$

$$+ O\left(\sqrt{\frac{L}{N}}\right) + O\left(\sqrt{\frac{\log d}{N}}\right) + O\left(\sqrt{\frac{\log 1/\omega}{N}}\right) + O\left(\frac{q\epsilon_p}{\sqrt{d}}\right) + O\left(\epsilon_w \sqrt{\frac{q \log 1/\omega}{d}}\right),$$

*where $\tilde{S} = S|_{\mathcal{P}} + \delta^w$ is the watermarked dataset, $\eta \sim \mathcal{U}\{-\epsilon_w, \epsilon_w\}^q$ is a random vector.*

After obtaining Corollary 5.7, similar to post-poisoning watermarking, we can ensure the soundness of poisoning-concurrent watermarking by the following corollary:

**Corollary 5.8** (Poisoning-concurrent watermarking). *If $\mathcal{D}'$ is $(\lambda, \mu)$-effective for some $\mu > 0$, when $N$ and $d$ are sufficiently large, $q = O\left(\sqrt{d}/\epsilon_p\right)$, with high probability, the network $\mathcal{F}$ is trained on a poisoning-concurrent watermarking dataset $\{x_i + \delta_i^p \oplus \delta^w, y_i\}_{i=1}^N$ that satisfies $\mathcal{R}^{\mathrm{poi}}\left(\mathcal{D}^{\mathrm{victim}}, \mathcal{F}\right) \leq \lambda$.*

**Comparison of two types of watermarking.** For post-poisoning watermarking, the total perturbation budget will become $\epsilon_w + \epsilon_p$. To ensure the detectability, the watermarking length is expected to be $\Theta(\sqrt{d}/\epsilon_w)$, and when ensuring the utility of poisoning, no additional requirement is needed. In comparison, for poisoning-concurrent watermarking, the total perturbation budget is $\max\{\epsilon_w, \epsilon_p\}$, which is smaller than the post-poisoning case $\epsilon_w + \epsilon_p$. The watermarking length needed to guarantee the detectability becomes looser, $\Theta(1/\epsilon_w^2)$, but the poisoning utility requires a larger $O\left(\sqrt{d}/\epsilon_p\right)$. We will verify these results in Section 6.

# 6 Experiments

## 6.1 Experimental setup

**Baseline methods.** We evaluate our approach using two imperceptible clean-label backdoor attacks, Narcissus [91] and AdvSc [88], as well as two imperceptible clean-label availability attacks, UE [37] and AP [22]. We evaluate on CIFAR-10, CIFAR-100 [45], and Tiny-ImageNet dataset [48]. The accuracy and attack success rate are measured on various victim models including ResNet-18, ResNet-50 [33], VGG-19 [72], DenseNet121 [36], WRN34-10 [90], MobileNet v2 [68].

**Implementation details.** We apply both post-poisoning watermarking and poisoning-concurrent watermarking to craft watermarks for each method. The watermarking algorithms are shown in Appendix C. We evaluate watermarking lengths ranging from 0 to 3000, randomly select the watermarking dimensions while fixing the random seed to ensure reproducibility. The watermarking and poisoning budgets are set to $16/255$ for backdoor attacks, and $8/255$ for availability attacks. For victim model training, the total epochs are 200, initial learning rate is 0.5 with a cosine scheduler, the momentum and weight decay are 0.9 and $10^{-4}$ respectively.

## 6.2 Main Results

Tables 1 and 2 present the evaluation results of watermarking on backdoor and availability attacks respectively. The results show that as the watermarking length $q$ increases, the detection performance (quantified by the AUROC score) improves consistently, achieving perfect detection (i.e., AUROC score be 1) when $q$ is sufficiently large. This confirms the theoretical findings in Section 4, which state that when $q$ exceeds a certain threshold ($\Theta(\sqrt{d}/\epsilon_w)$ for post-poisoning and $\Theta(1/\epsilon_w^2)$ for poisoning-concurrent), the watermarking provides provable and reliable detectability. Furthermore, poisoning-concurrent watermarking consistently outperforms post-poisoning watermarking for the same $q$, corroborating Remark 4.5, which indicates that $\Omega(1/\epsilon_w^2)$ is smaller than $\Omega(\sqrt{d}/\epsilon_w)$.

We also evaluate the poisoning performance under watermarking, measured by test accuracy and attack success rate (ASR) for backdoor attacks, and test accuracy for availability attacks. The results indicate that, for post-poisoning watermarking, all four attacks demonstrate strong performance compared to baseline methods without watermarking, supporting Theorem 5.2, which asserts that post-poisoning watermarking preserves poisoning when $d$ and $N$ are sufficiently large, regardless of $q$. For AdvSc, the ASR slightly decreases when $q$ is large. This may be attributed to the reliance of AdvSc on shortcuts in the left-top $1/4$ dimension [88], implicitly reducing the effective poison dimension to $\frac{1}{4}d$ and weakening its poisoning effect. Despite this limitation, AdvSc still achieves a respectable ASR of approximately 80%. For poisoning-concurrent watermarking, the ASR for backdoor attacks and the test accuracy drop for availability attacks are more sensitive to watermarking length $q$. Specifically, for Narcissus and AdvSc, the ASR drops below 30% when $q$ reaches 2000. For UE and AP, test accuracy recovers to about 90% when $q$ reaches 2500 and 3000 respectively, rendering the poisoning ineffective. These observations align with Theorem 5.6 and Corollaries 5.7 and 5.8, which emphasize that maintaining poisoning effectiveness in poisoning-concurrent watermarking requires $q$ to remain below $O(\sqrt{d}/\epsilon_p)$. When $q$ exceeds this bound, the watermarking begins to dominate, significantly reducing poisoning efficacy.

For experimental results under more datasets and network structures, please refer to Appendix D.

## 6.3 Ablation Studies

Table 2: The clean accuracy (Acc,%) and AUROC of UE and AP availability attacks both on post-poisoning watermarking and poisoning-concurrent watermarking with different watermarking length $q$ under ResNet-18 and CIFAR-10.

| Length/Method | UE [37] | | AP [22] | |
|---|---|---|---|---|
| Acc($\downarrow$)/AUROC($\uparrow$) | Post-Poisoning | Poisoning-Concurrent | Post-Poisoning | Poisoning-Concurrent |
| 0(Baseline) | 10.79/- | 10.79/- | 8.53/- | 8.53/- |
| 100 | 10.03/0.5844 | 10.35/0.8197 | 10.14/0.5688 | 10.30/0.6950 |
| 300 | 11.45/0.7067 | 9.70/0.9684 | 10.08/0.7573 | 11.77/0.7732 |
| 500 | 11.71/0.7810 | 10.02/0.9930 | 8.71/0.8623 | 15.84/0.8931 |
| 1000 | 11.37/0.9499 | 9.42/0.9991 | 10.58/0.9742 | 21.87/0.9949 |
| 1500 | 9.94/0.9786 | 10.10/0.9997 | 11.02/0.9916 | 32.46/0.9995 |
| 2000 | 9.06/0.9992 | 10.03/1.0000 | 10.48/0.9987 | 38.62/1.0000 |
| 2500 | 10.44/0.9996 | 88.78/1.0000 | 12.68/1.0000 | 36.79/1.0000 |
| 3000 | 9.99/1.0000 | 91.79/1.0000 | 13.52/1.0000 | 93.40/1.0000 |

**Watermarking budget.** We analyze the impact of watermarking budget $\epsilon_w$ on poisoning-concurrent watermarking for AdvSc attack. The results presented in Figure 1 show that as the budget increases, the detection performance (AUROC) improves. This observation verifies Theorem 4.9, which states that a larger $\epsilon_w$ allows for a smaller $q$ to achieve effective detection. However, the poisoning performance (ASR) decreases as $\epsilon_w$ grows, confirming Corollary 5.7, which suggests that larger $\epsilon_w$ results in a higher risk $\mathcal{R}(\mathcal{D}', \mathcal{F})$, thereby degrading the poisoning power. More results are provided in Appendix D.3.

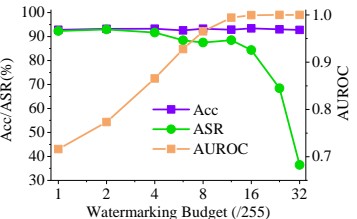

Figure 1: The Acc, ASR and AUROC of AdvSc backdoor attack on different budget $\epsilon_w$ for poisoning-concurrent watermarking with $q = 1000$.

**Position of watermarking dimension.** Our theoretical guarantees indicate that the position of the watermarking dimensions $\mathcal{W}$ has no significant impact. By default, we set $\mathcal{W}$ to be randomly selected from $[d]$. To validate this, we test fixed watermarking positions on the left-top (LT), left-bottom (LB), right-top (RT) and right-bottom (RB) regions of the image. We conduct experiments on post-poisoning UE watermarking with a length of 500. Results shown in Figure 2 demonstrate that the position of watermarking dimensions has minimal impact for both detection and poisoning performance.

In Appendix E, we have further discussed potential defense and watermark removal methods, including data augmentations, image regeneration attacks, differential privacy noises, and diffusion purification.

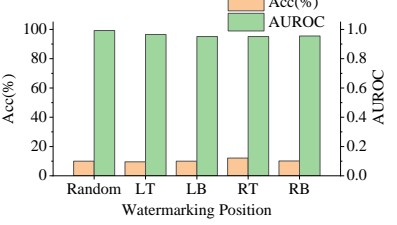

Figure 2: The Acc and AUROC of UE availability attack on different watermarking position for poisoning-concurrent watermarking with $q = 500$.

## 7 Conclusion

In this paper, we propose two provable and practical watermarking methods for data poisoning attacks: post-poisoning watermarking and poisoning-concurrent watermarking. We provide theoretical guarantees for the soundness of these watermarking methods, certifying their effectiveness when the watermarking length is $\Theta(\sqrt{d}/\epsilon_w)$ and $\Theta(1/\epsilon_w^2)$ for post-poisoning and poisoning-concurrent watermarking. Furthermore, we prove the soundness of the poisoning of post-poisoning and poisoning-concurrent watermarking when the length is $O(\sqrt{d}/\epsilon_p)$. We validate our theoretical findings through evaluation on several data poisoning attacks, including backdoor and availability attacks.

**Limitation and future works.** While our watermarking methods offer sufficient conditions for both detection and poisoning utility, the necessary conditions for these properties remain an open area for future research. Moreover, exploring more sophisticated watermarking designs that could achieve better performance and robustness in both detection and poisoning utility is a promising direction for further development.

## Acknowledgment

This paper is supported by the Strategic Priority Research Program of CAS Grant XDA0480502, the Robotic AI-Scientist Platform of Chinese Academy of Sciences, NSFC Grants 12288201 and 92270001, and the CAS Project for Young Scientists in Basic Research Grant YSBR-040.

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

# Appendix

## A  Symbol Table

| Notation | Description |
| --- | --- |
| $d$ | The dimension of data |
| $q$ | The dimension of watermarking |
| $N$ | The number of samples in a dataset |
| $\mathcal{P}$ | The indices of poisoning dimension |
| $\mathcal{W}$ | The indices of watermarking dimension |
| $\epsilon_p$ | The perturbation budget of a poisoning attack |
| $\epsilon_w$ | The perturbation budget of a watermark |
| $\delta^p$ | A data poisoning attack |
| $\delta^w$ | A watermark |
| $\delta_x$ | A sample-wise perturbation on data $x$ |
| $\zeta$ | A key |
| $\Xi$ | A key distribution |
| $S$ | A clean dataset |
| $S'$ | A perturbed dataset |
| $\mathcal{D}$ | A clean data distribution |
| $\mathcal{D}'$ | A data distribution under some perturbations |
| $L$ | The layer of a neural network |
| $\mathcal{L}$ | A loss function |
| $\mathcal{F}$ | A model (neural network) |
| $\mathcal{R}$ | A generalization risk |
| $\mathcal{R}^{\text{poi}}$ | A poisoning objective risk |
| $\omega$ | A probability |

## B  Proofs

### B.1  Proofs of Theorems in Section 4.1

**Lemma B.1** (McDiarmid's Inequality [56]). *Let $X_1, X_2, \cdots, X_n$ be independent random variables on $\mathcal{X}_1, \mathcal{X}_2, \cdots, \mathcal{X}_n$ and $f : \mathcal{X}_1 \times \mathcal{X}_2 \times \cdots \times \mathcal{X}_n \to \mathbb{R}$ be a multivariate function. If there exist positive constants $c_1, c_2, \cdots, c_n$, such that for all $(x_1, x_2, \cdots, x_n) \in \mathcal{X}_1 \times \mathcal{X}_2 \times \cdots \times \mathcal{X}_n$ and $i \in [n]$, it has*

$$\sup_{x_i' \in \mathcal{X}_i} |f(x_1, \cdots, x_{i-1}, x_i', x_{i+1}, \cdots, x_n) - f(x_1, \cdots, x_{i-1}, x_i, x_{i+1}, \cdots, x_n)| \le c_i,$$

*then for any $\epsilon > 0$, the following inequalities hold*

$$\mathbb{P}(f(X_1, X_2, \cdots, X_n) - \mathbb{E}[f(X_1, X_2, \cdots, X_n)] \ge \epsilon) \le e^{-\frac{2\epsilon^2}{\sum_{i=1}^n c_i^2}},$$

$$\mathbb{P}(f(X_1, X_2, \cdots, X_n) - \mathbb{E}[f(X_1, X_2, \cdots, X_n)] \le -\epsilon) \le e^{-\frac{2\epsilon^2}{\sum_{i=1}^n c_i^2}}.$$

**Definition B.2** (Random identical key). The random identical key means that for each entry, the probability of its value being 1 or $-1$ is $1/2$, i.e., $\zeta^i = \mathcal{U}\{-1, +1\}$ for all entries $i$ of key $\zeta$.

**Theorem B.3** (Theorem 4.1, restated). *For any data point $x$ sampled from $\mathcal{D}_\mathcal{X}$ and their corresponding poison be $\delta_x^p$, there exists a distribution $\Xi$ defined in $\mathbb{R}^d$ such that we can sample the key $\zeta \sim \Xi$ satisfied that for any $\omega \in (0,1)$, there are:*

*(1): $\mathbb{P}_{x \sim \mathcal{D}_\mathcal{X}, \zeta \sim \Xi}\left(\zeta^T x < \sqrt{\frac{d}{2} \log \frac{1}{\omega}}\right) > 1 - w$; (2): we can craft the watermark $\delta_x^w$ based on $\zeta$ such that $\mathbb{P}_{x \sim \mathcal{D}_\mathcal{X}, \zeta \sim \Xi}\left(\zeta^T(x + \delta_x) > q\epsilon_w - \sqrt{\frac{d}{2} \log \frac{1}{\omega}}\right) > 1 - w$. Hence, when $q > \frac{1}{\epsilon_w}\sqrt{2d \log \frac{1}{\omega}}$, it holds that $\mathbb{P}_{x_1, x_2 \sim \mathcal{D}_\mathcal{X}, \zeta \sim \Xi}\left(\zeta^T(x_1 + \delta_1) > \zeta^T x_2\right) > 1 - 2\omega$.*

*Proof of Theorem 4.1.* (1): Denote the distribution $\Xi$ be the distribution of a random identical key, i.e., $\Xi = \mathcal{U}\{-1, +1\}^d$, it has

$$\mathbb{E}_\zeta[\zeta^T x] = 0$$

for all $x \in \mathcal{D}$. Furthermore, as $x$ lies in $[0,1]$, it always holds that

$$|\zeta^i x^i - \zeta^i \tilde{x}^i| \leq |\zeta^i| = 1$$

for all $i$, $\zeta$ and $x, \tilde{x} \in \mathcal{D}$.

Therefore, by McDiarmid's inequality, for any $\alpha > 0$, it has

$$\mathbb{P}_x \left[ \mathbb{P}_\zeta [\zeta^T x \geq \alpha] \leq e^{-\frac{2\alpha^2}{d}} \right] = 1,$$

which concludes that

$$\mathbb{P}_{x,\zeta}[\zeta^T x \geq \alpha] \leq e^{-\frac{2\alpha^2}{d}}.$$

Therefore, let $\omega = e^{-\frac{2\alpha^2}{d}}$, it has $\alpha = \sqrt{\frac{d}{2} \log \frac{1}{\omega}}$, which validates (1).

(2): For any key $\zeta$, we craft the watermark $\delta_x^w$ as $(\epsilon_w \cdot \zeta^{d_i})_{i=1}^q$. We can conclude that

$$\mathbb{E}_\zeta[\zeta^T(x + \delta_x)] = \mathbb{E}_\zeta[\zeta^T x] + \mathbb{E}_\zeta[\zeta^T \delta_x^p] + \mathbb{E}_\zeta[\zeta^T \delta_x^w].$$

Because $x$ and $\delta_x^p$ are independent from $\zeta$, it holds that

$$\mathbb{E}_\zeta[\zeta^T x] = \mathbb{E}_\zeta[\zeta^T \delta_x^p] = 0.$$

Therefore, we have

$$\mathbb{E}_\zeta[\zeta^T(x + \delta_x)] = \mathbb{E}_\zeta[\zeta^T \delta_x^w] = q\epsilon_w.$$

Similar to (1), by McDiarmid's inequality, for any $\beta > 0$, it has

$$\mathbb{P}_{x,\zeta}[\zeta^T(x + \delta_x) - q\epsilon_w \leq -\beta] \leq e^{-\frac{2\beta^2}{d}}.$$

Therefore, let $\omega = e^{-\frac{2\beta^2}{d}}$, it has $\beta = \sqrt{\frac{d}{2} \log \frac{1}{\omega}}$, which induces that

$$\mathbb{P}_{x \sim \mathcal{D}_\mathcal{X}, \zeta} \left( \zeta^T(x + \delta_x) > q\epsilon_w - \sqrt{\frac{d}{2} \log \frac{1}{\omega}} \right) > 1 - w.$$

When $q > \frac{1}{\epsilon_w} \sqrt{2d \log \frac{1}{\omega}}$, it holds that

$$\sqrt{\frac{d}{2} \log \frac{1}{\omega}} < q\epsilon_w - \sqrt{\frac{d}{2} \log \frac{1}{\omega}}.$$

Hence by the union bound, it has

$$\mathbb{P}_{x_1, x_2 \sim \mathcal{D}_\mathcal{X}, \zeta} \left[ \zeta^T(x_1 + \delta_1) > \zeta^T x_2 \right] = 1 - \mathbb{P}_{x_1, x_2 \sim \mathcal{D}_\mathcal{X}, \zeta} \left[ \zeta^T(x_1 + \delta_1) \leq \zeta^T x_2 \right]$$

$$\geq 1 - \mathbb{P}_{x_1, x_2 \sim \mathcal{D}_\mathcal{X}, \zeta} \left[ \zeta^T(x_1 + \delta_1) \leq q\epsilon_w - \sqrt{\frac{d}{2} \log \frac{1}{\omega}} \right] - \mathbb{P}_{x_1, x_2 \sim \mathcal{D}_\mathcal{X}, \zeta} \left[ \zeta^T x_2 \geq \sqrt{\frac{d}{2} \log \frac{1}{\omega}} \right]$$

$$\geq 1 - 2\omega. \tag{1}$$

$\square$

**Theorem B.4** (Theorem 4.3, restated). *For any $x \sim \mathcal{D}_\mathcal{X}$, there exists a distribution $\Xi \in \mathbb{R}^d$ such that we can sample the key $\zeta \sim \Xi$ satisfied that for any $\omega \in (0,1)$:*

*(1): $\mathbb{P}_{x \sim \mathcal{D}_\mathcal{X}, \zeta \sim \Xi} \left( \zeta^T x < \sqrt{\frac{q}{2} \log \frac{1}{\omega}} \right) > 1 - w$; (2): we can craft the watermark $\delta_x^w$ and poison $\delta_x^p$ such that $\mathbb{P}_{x \sim \mathcal{D}_\mathcal{X}, \zeta \sim \Xi} \left( \zeta^T(x + \delta_x) > q\epsilon_w - \sqrt{\frac{q}{2} \log \frac{1}{\omega}} \right) > 1 - w$. Hence, when $q > \frac{2}{\epsilon_w^2} \log \frac{1}{\omega}$, it holds that $\mathbb{P}_{x_1, x_2 \sim \mathcal{D}_\mathcal{X}, \zeta \sim \Xi} \left( \zeta^T(x_1 + \delta_1) > \zeta^T x_2 \right) > 1 - 2\omega$.*

*Proof of Theorem 4.3.* For poisoning-concurrent watermarking, denote the poisoning dimension be $\mathcal{P}$ and the watermarking dimension be $\mathcal{W}$, where $[d] = \mathcal{P} \cup \mathcal{W}$ and $|\mathcal{W}| = q$.

We sample the key $\zeta \in \mathbb{R}^d$ from a certain distribution $\Xi$, such that $\zeta^i = \mathcal{U}\{-1, +1\}, i \in \mathcal{W}$ and $\zeta^i = 0, i \in \mathcal{P}$.

Therefore, by McDiarmid's inequality, for any $\alpha > 0$, it has

$$\mathbb{P}_{x,\zeta}[\zeta^T x \geq \alpha] \leq e^{-\frac{2\alpha^2}{q}}.$$

Let $\omega = e^{-\frac{2\alpha^2}{q}}$, it has $\alpha = \sqrt{\frac{q}{2}\log\frac{1}{\omega}}$, which validates condition (1).

We craft the watermark $\delta_x^w$ as $(\epsilon_w \cdot \zeta^i)_{i=1}^d$. Similar to the proof of Theorem 4.1, we can conclude that

$$\mathbb{E}_\zeta[\zeta^T(x + \delta_x)] = \mathbb{E}_\zeta[\zeta^T \delta_x^w] = q\epsilon_w.$$

Let $\omega = e^{-\frac{2\beta^2}{q}}$, it has $\beta = \sqrt{\frac{q}{2}\log\frac{1}{\omega}}$, which induces that

$$\mathbb{P}_{x \sim \mathcal{D}_\mathcal{X},\zeta}\left(\zeta^T(x + \delta_x) > q\epsilon_w - \sqrt{\frac{q}{2}\log\frac{1}{\omega}}\right) > 1 - w.$$

When $q > \frac{2}{\epsilon_w^2}\log\frac{1}{\omega}$, it holds that

$$\sqrt{\frac{q}{2}\log\frac{1}{\omega}} < q\epsilon_w - \sqrt{\frac{q}{2}\log\frac{1}{\omega}}.$$

Hence by union bound, it has

$$\mathbb{P}_{x_1,x_2 \sim \mathcal{D}_\mathcal{X},\zeta}\left[\zeta^T(x_1 + \delta_1) > \zeta^T x_2\right] \geq 1 - 2\omega.$$

$\square$

## B.2 Proofs of Theorems in Section 4.2

**Theorem B.5** (Proposition 4.7, restated). *For the dataset $S_\mathcal{X}$, when $q > \frac{2+\epsilon_p}{\epsilon_w}\sqrt{\frac{d}{2}\log\frac{2N}{\omega}}$, we can sample the key $\zeta \in \mathbb{R}^d$ from a certain distribution such that, with a probability of at least $1 - \omega$, there exists the watermark $\delta^w$ such that $\zeta^T(x_j + \delta_j) > \zeta^T x_i, \forall i, j \in [N]$.*

*Proof of Proposition 4.7.* For the random identical key $\zeta \in \mathbb{R}^d$, and $x_i \in [0,1]^d$, it holds that

$$|\zeta^j x_i^j - \zeta^j \tilde{x}_i^j| \leq |\zeta^j| = 1$$

for every $\tilde{x}_i \neq x_i$.

By McDiarmid's inequality, for any $\alpha > 0$. it has

$$\mathbb{P}_\zeta\left[\zeta^T x_i \geq \alpha\right] \leq e^{-\frac{2\alpha^2}{d}}.$$

Furthermore, as $\|\delta_i^p\| \leq \epsilon_p$, it has

$$|\zeta^j(\delta_i^p)^j - \zeta^j(\tilde{\delta}_i^p)^j| \leq |\zeta^j| \cdot \epsilon_p = \epsilon_p$$

for every $\tilde{\delta}_i^p \neq \delta_i^p$.

By McDiarmid's inequality, for any $\beta > 0$. it has

$$\mathbb{P}_\zeta\left[\zeta^T \delta_i^p \geq \beta\right] \leq e^{-\frac{2\beta^2}{d\epsilon_p^2}}.$$

By the union bound, it holds that

$$\mathbb{P}\left[\cup_{i=1}^N \{|\zeta^T x_i| \geq \alpha\}\right] \leq \sum_{i=1}^N \mathbb{P}\left[|\zeta^T x_i| \geq \alpha\right] \leq N e^{-\frac{2\alpha^2}{d}},$$

$$\mathbb{P}\left[\cup_{i=1}^N \{|\zeta^T \delta_i^p| \geq \beta\}\right] \leq \sum_{i=1}^N \mathbb{P}\left[|\zeta^T x_i| \geq \beta\right] \leq N e^{-\frac{2\beta^2}{d\epsilon_p^2}}.$$

We now craft the watermark $\delta^w$ such that

$$\delta^w = \epsilon_w \cdot \zeta|_{\mathcal{W}}.$$

It has

$$\zeta^T \delta^w = q\epsilon_w.$$

Therefore, let $\beta = \epsilon_p \alpha$, it holds that

$$
\begin{aligned}
\mathbb{P}\left[\cap_{i=1}^N \{\zeta^T(x_i + \delta^i) > q\epsilon_w - (1 + \epsilon_p)\alpha\}\right] &= \mathbb{P}\left[\cap_{i=1}^N \{\zeta^T(x_i + \delta_p^i + \delta^w) > q\epsilon_w - (1 + \epsilon_p)\alpha\}\right] \\
&= \mathbb{P}\left[\cap_{i=1}^N \{\zeta^T(x_i + \delta_p^i) > -(1 + \epsilon_p)\alpha\}\right] \\
&\geq \mathbb{P}\left[\{\cap_{i=1}^N \{\zeta^T \delta_p^i > -\epsilon_p \alpha\}\} \cap \{\cap_{i=1}^N \{\zeta^T x^i > -\alpha\}\}\right] \\
&\geq 1 - \mathbb{P}\left[\cup_{i=1}^N \{|\zeta^T x_i| \geq \alpha\}\right] - \mathbb{P}\left[\cup_{i=1}^N \{|\zeta^T \delta_i^p| \geq \epsilon_p \alpha\}\right] \\
&\geq 1 - 2Ne^{-\frac{2\alpha^2}{d}}.
\end{aligned}
$$

Therefore, when

$$q\epsilon_w - (1 + \epsilon_p)\alpha > \alpha,$$

it has

$$\zeta^T(x_j + \delta_j) > \zeta^T x_i, \forall i, j \in [N]$$

happens with probability at least $1 - 2Ne^{-\frac{2\alpha^2}{d}}$.

Let $\omega = 2Ne^{-\frac{2\alpha^2}{d}}$, the condition will be

$$q > \frac{2 + \epsilon_p}{\epsilon_w} \sqrt{\frac{d}{2} \log \frac{2N}{\omega}}.$$

$\square$

**Theorem B.6** (Theorem 4.9, restated). *For the dataset $S_{\mathcal{X}} = \{x_1, x_2, \cdots, x_N\}$, $x_i$ and the poison $\delta_p^i$ are i.i.d. sampled from $\mathcal{D}_{\mathcal{X}}$ and $\mathcal{D}_{\mathcal{P}}$ respectively. For any $w \in (0, 1/2)$ and $q > \frac{2}{\epsilon_w}\sqrt{2d \log \frac{1}{\omega}}$, we can sample the key $\zeta \in \mathbb{R}^d$ from a certain distribution such that, with probability at least $1 - 2\exp\left(\frac{-N\left(\omega - e^{-q^2 \epsilon_w^2/8d}\right)^2}{\omega + e^{-q^2 \epsilon_w^2/8d}}\right)$, it is possible to craft the watermark $\delta^w$, such that $\zeta^T(x_i + \delta_i) > \frac{q\epsilon_w}{2}, \zeta^T x_i < \frac{q\epsilon_w}{4}$ holds for at least $(1 - 2\omega)N$ samples.*

*Proof of Theorem 4.9.* Denote the failure cases of Proposition 4.7 be

$$F_i(\alpha) = \mathbb{I}\{\zeta^T x_i \geq \alpha\}, F_i'(\alpha) = \mathbb{I}\{\zeta^T \delta_i^p \geq \epsilon_p \alpha\}$$

and

$$F(\alpha) = \sum_{i=1}^N F_i(\alpha), F'(\alpha) = \sum_{i=1}^N F_i'(\alpha)$$

Due to the i.i.d property of $x_i$ and $\delta_i^p$, $F_i(\alpha)$ and $F_i'(\alpha)$ are also i.i.d. for $i = \{1, 2, \cdots, N\}$. $\zeta^T(x_i + \delta_i) > \zeta^T x_i$ holds as long as both $F_i(\alpha) = 0$ and $F_i'(\alpha) = 0$ for a certain constant $\alpha > 0$.

By McDiarmid's inequality,

$$\mathbb{P}[F_i(\alpha) = 1] \leq e^{-\frac{2\alpha^2}{d}}, \mathbb{P}[F_i'(\alpha) = 1] \leq e^{-\frac{2\alpha^2}{d}}.$$

Therefore, assume that

$$\mathbb{P}[F_i(\alpha) = 1] = p_{i,\alpha}, \mathbb{P}[F_i'(\alpha) = 1] = p_{i',\alpha}.$$

$F_i(\alpha)$ and $F_i'(\alpha)$ obey the Bernoulli distribution $\mathcal{B}(p_{i,\alpha})$ and $\mathcal{B}(p_{i',\alpha})$ respectively.

Denote $\bar{F}_i(\alpha)$ obeying the Bernoulli distribution $\mathcal{B}(e^{-\frac{2\alpha^2}{d}})$, and

$$\bar{F}(\alpha) = \sum_{i=1}^{N} \bar{F}_i(\alpha).$$

By the Chernoff bound, it holds that

$$\mathbb{P}\left[\bar{F}(\alpha) \geq (1+\delta)Ne^{-\frac{2\alpha^2}{d}}\right] \leq \exp\left(\frac{-\delta^2 N}{2+\delta}e^{-\frac{2\alpha^2}{d}}\right)$$

for any $\delta > 0$.

As it always has $\bar{F}_i(\alpha) \leq \bar{F}_i(\alpha), \bar{F}_i'(\alpha) \leq \bar{F}_i(\alpha)$, it holds that

$$\mathbb{P}\left[F(\alpha) \geq (1+\delta)Ne^{-\frac{2\alpha^2}{d}}\right] \leq \exp\left(\frac{-\delta^2 N}{2+\delta}e^{-\frac{2\alpha^2}{d}}\right),$$

$$\mathbb{P}\left[F'(\alpha) \geq (1+\delta)Ne^{-\frac{2\alpha^2}{d}}\right] \leq \exp\left(\frac{-\delta^2 N}{2+\delta}e^{-\frac{2\alpha^2}{d}}\right).$$

Let $\omega = (1+\delta)e^{-\frac{2\alpha^2}{d}}$. It has

$$\mathbb{P}\left[F(\alpha) \geq \omega N\right] \leq \exp\left(\frac{-N(\omega - e^{-2\alpha^2/d})^2}{\omega + e^{-2\alpha^2/d}}\right),$$

$$\mathbb{P}\left[F'(\alpha) \geq \omega N\right] \leq \exp\left(\frac{-N(\omega - e^{-2\alpha^2/d})^2}{\omega + e^{-2\alpha^2/d}}\right).$$

Therefore, the probability of a bad case is at most

$$\mathbb{P}[F(\alpha) \geq \omega N] + \mathbb{P}[F'(\alpha) \geq \omega N]$$

with $2\omega N$ samples. To achieve the non-vacuous gap of watermarking between poisoned data $x_i + \delta_i$ and benign data $x_j$, we can set

$$\alpha = \frac{q\epsilon_w}{4}.$$

In this case, if both $F_i(\alpha)$ and $F_i'(\alpha) = 0$, i.e., sample $x_i$ is not a bad case, it holds that

$$\zeta^T x_i < \alpha = \frac{q\epsilon_w}{4}, \zeta^T(x_i + \delta_i) = q\epsilon_w - (1+\epsilon_p)\alpha > \frac{q\epsilon_w}{2}.$$

Hence, for at least $(1 - 2\omega)N$ samples, with probability at least

$$1 - 2\exp\left(\frac{-N(\omega - e^{-2\alpha^2/d})^2}{\omega + e^{-2\alpha^2/d}}\right) = 1 - 2\exp\left(\frac{-N(\omega - e^{-q^2\epsilon_w^2/8d})^2}{\omega + e^{-q^2\epsilon_w^2/8d}}\right),$$

the property holds.

Furthermore, as we set

$$\omega = (1+\delta)e^{-\frac{2\alpha^2}{d}}$$

and $\delta > 0$. This condition is valid as long as

$$q > \frac{2}{\epsilon_w}\sqrt{2d\log\frac{1}{\omega}}$$

$\square$

**Theorem B.7** (Proposition 4.11, restated). *For the dataset $S_{\mathcal{X}}$, when $q > \frac{4}{\epsilon_w^2}\log\frac{N}{\omega}$, it is possible to sample the key $\zeta \in \mathbb{R}^d$ from a certain distribution such that, with probability at least $1 - \omega$, we can craft a watermark $\delta^w$ and poison $\delta^p$ such that $\zeta^T(x_j + \delta_j) > \zeta^T x_i, \forall i, j \in [N]$.*

*Proof of Proposition 4.11.* We sample the key $\zeta \in \mathbb{R}^d$, such that

$$\zeta^i = \mathcal{U}\{-1, +1\}, i \in \mathcal{W}$$

and

$$\zeta^i = 0, i \in \mathcal{P}.$$

By McDiarmid's inequality, for any $\alpha > 0$, it holds that

$$\mathbb{P}_\zeta \left[ \zeta^T x_i \geq \alpha \right] \leq e^{-\frac{2\alpha^2}{q}}.$$

By the union bound, it holds that

$$\mathbb{P} \left[ \cup_{i=1}^N \{ |\zeta^T x_i| \geq \alpha \} \right] \leq \sum_{i=1}^N \mathbb{P} \left[ |\zeta^T x_i| \geq \alpha \right] \leq N e^{-\frac{2\alpha^2}{q}}.$$

We now craft the watermark $\delta^w$ such that

$$\delta^w = \epsilon_w \cdot \zeta.$$

It holds that

$$\zeta^T \delta^w = q\epsilon_w.$$

It holds that

$$\begin{aligned}
\mathbb{P} \left[ \cap_{i=1}^N \{ \zeta^T (x_i + \delta^i) > q\epsilon_w - \alpha \} \right] &= \mathbb{P} \left[ \cap_{i=1}^N \{ \zeta^T x_i > q\epsilon_w - \alpha \} \right] \\
&= \mathbb{P} \left[ \cap_{i=1}^N \{ \zeta^T x_i > -\alpha \} \right] \\
&\geq 1 - \mathbb{P} \left[ \cup_{i=1}^N \{ |\zeta^T x_i| \geq \alpha \} \right] \\
&\geq 1 - N e^{-\frac{2\alpha^2}{q}}.
\end{aligned}$$

Therefore, when

$$q\epsilon_w > 2\alpha,$$

it holds that

$$\zeta^T (x_j + \delta_j) > \zeta^T x_i, \forall i, j \in [N]$$

happens with probability at least $1 - N e^{-\frac{2\alpha^2}{q}}$.

Let $\omega = N e^{-\frac{2\alpha^2}{q}}$, it holds that

$$\alpha = \sqrt{\frac{q}{2} \log \frac{N}{\omega}}.$$

Then the condition will be

$$q > \frac{4}{\epsilon_w^2} \log \frac{N}{\omega}.$$

$\square$

**Theorem B.8** (Theorem 4.13, restated). *For the dataset $S_\mathcal{X} = \{x_1, x_2, \cdots, x_N\}$, where $x_i$ is i.i.d. sampled from $\mathcal{D}_\mathcal{X}$. For any $\omega \in (0, 1)$ and $q > \frac{9}{2\epsilon_w^2} \log \frac{1}{\omega}$, it is possible to sample the key $\zeta \in \mathbb{R}^d$ from a certain distribution such that, with probability at least $1 - \exp\left(\frac{-N\left(\omega - e^{-2q\epsilon^2/9}\right)^2}{\omega + e^{-2q\epsilon^2/9}}\right)$, we can craft the watermark and the poison satisfies $\zeta^T (x_i + \delta_i) > \frac{2q\epsilon_w}{3}, \zeta^T x_i < \frac{q\epsilon_w}{3}$ holds for at least $(1 - \omega)N$ samples.*

*Proof of Theorem 4.13.* We sample the key $\zeta \in \mathbb{R}^d$, such that

$$\zeta^i = \mathcal{U}\{-1, +1\}, i \in \mathcal{W}$$

and
$$\zeta^i = 0, i \in \mathcal{P}.$$

Similar to the proof of 4.9, denote the failure case
$$F_i(\alpha) = \mathbb{I}\{\zeta^T x_i > \alpha\}$$
and
$$F(\alpha) = \sum_{i=1}^{N} F_i(\alpha).$$

By McDiarmid's inequality,
$$\mathbb{P}[F_i(\alpha) = 1] \le e^{-\frac{2\alpha^2}{q}}.$$

Denote $\bar{F}_i(\alpha)$ obeys the Bernoulli distribution $\mathcal{B}\left(e^{-\frac{2\alpha^2}{q}}\right)$, and
$$\bar{F}(\alpha) = \sum_{i=1}^{N} \bar{F}_i(\alpha).$$

By Chernoff bound, it holds that
$$\mathbb{P}\left[\bar{F}(\alpha) \ge (1+\delta)Ne^{-\frac{2\alpha^2}{q}}\right] \le \exp\left(\frac{-\delta^2 N}{(2+\delta)}e^{-\frac{2\alpha^2}{q}}\right)$$
for any $\delta > 0$.

As it always has $\bar{F}_i(\alpha) \le \bar{F}_i(\alpha)$, it holds that
$$\mathbb{P}\left[F(\alpha) \ge (1+\delta)Ne^{-\frac{2\alpha^2}{q}}\right] \le \exp\left(\frac{-\delta^2 N}{(2+\delta)}e^{-\frac{2\alpha^2}{q}}\right).$$

Let $\omega = (1+\delta)e^{-\frac{2\alpha^2}{q}}$. It has
$$\mathbb{P}\left[F(\alpha) \ge \omega N\right] \le \exp\left(\frac{-N(\omega - e^{-2\alpha^2/q})^2}{\omega + e^{-2\alpha^2/q}}\right),$$

Therefore, the probability of a bad case is at most
$$\mathbb{P}[F(\alpha) \ge \omega N]$$
with $\omega N$ samples. To achieve the non-vacuous gap of watermarking between poisoned data $x_i + \delta_i$ and benign data $x_j$, we can set
$$\alpha = \frac{q\epsilon_w}{3}.$$

In this case, if both $F_i(\alpha) = 0$, i.e., sample $x_i$ is not a bad case, it holds that
$$\zeta^T x_i \le \alpha = \frac{q\epsilon_w}{3}, \zeta^T(x_i + \delta_i) = q\epsilon_w - \alpha \ge \frac{2q\epsilon_w}{3}.$$

Hence for at least $(1-\omega)N$ samples, with probability at least
$$1 - \exp\left(\frac{-N(\omega - e^{-2\alpha^2/q})^2}{\omega + e^{-2\alpha^2/q}}\right) = 1 - \exp\left(\frac{-N(\omega - e^{-2q\epsilon^2/9})^2}{\omega + e^{-2q\epsilon^2/9}}\right),$$
the property holds.

Furthermore, as we set
$$\omega = (1+\delta)e^{-\frac{2\alpha^2}{q}}$$
and $\delta > 0$. This condition is valid as long as
$$q > \frac{9}{2\epsilon_w^2}\log\frac{1}{\omega}$$

$\square$

**Theorem B.9** (Theorem 4.15, restated). *For the dataset $S_{\mathcal{X}} = \{x_1, x_2, \cdots, x_N\}$, $x_i$ and the poison $\delta_p^i$ are i.i.d. sampled from $\mathcal{D}_{\mathcal{X}}$ and $\mathcal{D}_{\mathcal{P}}$ respectively. Considering a universal watermark $\delta^w$, with probability at least $1 - 2\mu$ for the sampled data and poisons, if there exists a key $\zeta$ that satisfies $\zeta^T(x_i + \delta_i) > C_1, \zeta^T x_i < C_2$, for at least $(1 - \omega)N$ samples $x_i$, then it holds that*

$$\mathbb{P}_{x,\tilde{x} \sim \mathcal{D}_{\mathcal{X}}, \delta^p \sim \Delta} \left( \{\zeta^T(x + \delta^p + \delta^w) > C_1, -\zeta^T\tilde{x} < C_2\} \right)$$

$$> 1 - 2\omega - 2\sqrt{\frac{d}{N}(\log(\frac{2N}{d}) + 1) - \frac{1}{N}\log(\frac{\mu}{4})}.$$

**Lemma B.10** (VC bound [81]). *Let $S = \{(x_i, y_i)\}_{i=1}^N$ be the training dataset, $(x_i, y_i) \sim \mathcal{D}$, where $\mathcal{D}$ is the data distribution. Then with probability at least $1 - \delta$, it holds that*

$$\mathbb{E}_{(x,y)\sim\mathcal{D}}\mathcal{L}(f(x), y) \leq \frac{1}{N}\sum_{i=1}^N \mathcal{L}(f(x_i), y_i) + \sqrt{\frac{VC(f)}{N}\left(\log\left(\frac{2N}{VC(f)}\right) + 1\right) - \frac{1}{N}\log\left(\frac{\delta}{4}\right)},$$

*where $VC(f)$ is the VC-dimension of the classifier $f$, $L(\cdot)$ is the loss function.*

**Lemma B.11** (VC-dimension of linear classifier). *The VC-dimension of linear classifiers $f_\theta = \{x \rightarrow 2\mathbb{I}(\theta^T x \geq 0) - 1; \theta \in \mathbb{R}^d\}$ is $d$.*

*Proof of Lemma B.11.* We need to prove that $f_\theta$ can shatter $d$ points and cannot shatter $d + 1$ points.

To prove that $f_\theta$ can shatter $d$ points, we only need to prove that $f_\theta$ can shatter $x_j = e_j, j \in [d]$ where $e_j$ is the basis of the space $\mathbb{R}^d$. In fact, for every $y_j \in \{-1, +1\}$, we can let

$$\theta = \sum_{j=1}^d y_j \cdot e_j.$$

Then it holds that

$$2\mathbb{I}(\theta^T x_j \geq 0) - 1 = y_j$$

for all $j \in [d]$.

Then we prove that $d + 1$ points cannot be shattered. We consider points $\{x_j\}_{j=1}^{d+1}$. Because $x_j \in \mathbb{R}^d$, $\{x_j\}_{j=1}^{d+1}$ are linearly dependent. Without loss of generality, we can assume that

$$x_{d+1} = \sum_{j=1}^d k_j x_j.$$

Now we can craft labels $\{y_j\}_{j=1}^{d+1}$ such that for any $f_\theta$, there exists

$$f_\theta(x_j) = 2\mathbb{I}(\theta^T x_j \geq 0) - 1 \neq y_j.$$

For $k_j \neq 0$, we set $y_j = 2\mathbb{I}(k_j \geq 0) - 1$, and we set $y_{d+1} = -1$. In this case, if the classifier $f_\theta$ can correctly classify $x_1, \cdots, x_d$, it must have

$$2\mathbb{I}(\theta^T x_j \geq 0) - 1 = y_j = 2\mathbb{I}(k_j \geq 0) - 1.$$

Therefore, $\mathbb{I}(\theta^T x_j \geq 0) = \mathbb{I}(k_j \geq 0)$. However, for $x_{d+1}$, it has

$$\theta^T x_{d+1} = \sum_{j=1}^d k_j \theta^T x_j \geq 0,$$

making

$$2\mathbb{I}(\theta^T x_{d+1} \geq 0) = +1 \neq y_{d+1}.$$

Therefore, $d + 1$ points cannot be shattered, resulting in $VC(f_\theta) = d$. $\square$

*Proof of Theorem 4.15.* Denote the classifier $h_1$ and $h_2$ as

$$h_1(x) = 2\mathbb{I}(\zeta^T x > C_1) - 1, h_2(x) = 2\mathbb{I}(\zeta^T x < C_2) - 1.$$

Denote the loss function $L_{0-1}$ as the 0-1 loss.

By Lemmas B.10 and B.11, with probability at least $1 - \mu$, it holds that

$$\mathbb{E}_{(x+\delta,+1)} L_{0-1}\left(h_1(x+\delta),+1\right) \leq \frac{1}{N}\sum_{i=1}^{N} L_{0-1}\left(h_1(x_i+\delta_i),+1\right) + \sqrt{\frac{d}{N}\left(\log\left(\frac{2N}{d}\right)+1\right) - \frac{1}{N}\log\left(\frac{\mu}{4}\right)}$$

$$= \frac{1}{N}\sum_{i=1}^{N} \mathbb{I}(\zeta^T(x_i+\delta_i) \leq C_1) + \sqrt{\frac{d}{N}\left(\log\left(\frac{2N}{d}\right)+1\right) - \frac{1}{N}\log\left(\frac{\mu}{4}\right)}$$

$$\leq \omega + \sqrt{\frac{d}{N}\left(\log\left(\frac{2N}{d}\right)+1\right) - \frac{1}{N}\log\left(\frac{\mu}{4}\right)}.$$

Similarly, it has

$$\mathbb{E}_{(x,-1)} L_{0-1}\left(h_2(x),-1\right) \leq \omega + \sqrt{\frac{d}{N}\left(\log\left(\frac{2N}{d}\right)+1\right) - \frac{1}{N}\log\left(\frac{\mu}{4}\right)}$$

Therefore,

$$\mathbb{P}_{x_1,x_2 \sim \mathcal{D}_{\mathcal{X}}, \delta_1 \sim \Delta}\left(\left\{\zeta^T(x_1+\delta_1) > C_1, -\zeta^T x_2 < C_2\right\}\right)$$

$$\geq 1 - \mathbb{P}_{x_1 \sim \mathcal{D}_{\mathcal{X}}, \delta_1 \sim \Delta}\left(\zeta^T(x_1+\delta_1) \leq C_1\right) - \mathbb{P}_{x_2 \sim \mathcal{D}_{\mathcal{X}}}(\zeta^T x_2 \geq C_2)$$

$$= 1 - \mathbb{E}_{(x+\delta,+1)} L_{0-1}\left(h_1(x+\delta),+1\right) - \mathbb{E}_{(x,-1)} L_{0-1}\left(h_2(x),-1\right)$$

$$\geq 1 - 2\omega - 2\sqrt{\frac{d}{N}\left(\log\left(\frac{2N}{d}\right)+1\right) - \frac{1}{N}\log\left(\frac{\mu}{4}\right)}.$$

$\square$

## B.3 Proofs of Theorems in Section 5

**Theorem B.12** (Theorem 5.2, restated). *With probability at least $1 - 2\omega$ for the poisoned dataset $\{(x_i', y_i)\}_{i=1}^{N} = S' \sim \mathcal{D}'$ and the key $\zeta \in \mathbb{R}^d$ selected from a certain distribution, we can craft the watermark $\delta^w$ satisfied:*

$$\mathcal{R}(\mathcal{D}', \mathcal{F}_{S'+\delta^w}^*) \leq \mathbb{E}_\eta \frac{1}{N}\sum_{i=1}^{N} \mathcal{L}(\mathcal{F}_{S'}^*(x_i'+\eta), y_i) + O\left(\sqrt{\frac{L}{N}}\right)$$

$$+ O\left(\sqrt{\frac{\log d}{N}}\right) + O\left(\sqrt{\frac{\log 1/\omega}{N}}\right) + O\left(\epsilon_w \sqrt{\frac{q\log 1/\omega}{d}}\right),$$

*where $S' + \delta^w = \{(x_i' + \delta^w, y_i)\}_{i=1}^{N}$ is the watermarked dataset, $\eta \sim \mathcal{U}\{-\epsilon_w, \epsilon_w\}^q$ is a random vector.*

*Proof of Theorem 5.2.* Let $x_i' = x_i + \delta_i^p$. For any random identical key $\zeta$, we craft the watermark $\delta^w$ as $(\epsilon_w \cdot \zeta^{d_i})_{i=1}^{q}$, which obey the distribution $\mathcal{U}\{-\epsilon_w, \epsilon_w\}^q$.

We first prove that

$$\frac{1}{N}\sum_{i=1}^{N} \mathcal{L}\left(\mathcal{F}_{S'+\delta^w}^*(x_i'), y_i\right) \leq \mathbb{E}_\eta \frac{1}{N}\sum_{i=1}^{N} \mathcal{L}\left(\mathcal{F}_{S'}^*(x_i'+\eta), y_i\right) + O\left(\epsilon_w \sqrt{\frac{q\log 1/\omega}{d}}\right).$$

Let $a = \min_{\mathcal{F}} \frac{1}{N}\sum_{i=1}^{N} L(\mathcal{F}(x_i'), y_i)$, the optimal classifier

$$\mathcal{F}_{S'+\delta^w}^*(t) = \arg\min_{\mathcal{F}} \frac{1}{N}\sum_{i=1}^{N} \mathcal{L}\left(\mathcal{F}(x_i'+\delta^w), y_i\right), \mathcal{F}_{S'}^*(t) = \arg\min_{\mathcal{F}} \frac{1}{N}\sum_{i=1}^{N} \mathcal{L}(\mathcal{F}(x_i'), y_i).$$

Let $G^* = \mathcal{F}^*_{S'}(t - \delta^w)$, it holds that

$$\frac{1}{N}\sum_{i=1}^{N}\mathcal{L}(G^*(x'_i + \delta^w), y_i) = \frac{1}{N}\sum_{i=1}^{N}\mathcal{L}(\mathcal{F}^*_{S'}(x'_i), y_i) = a.$$

Therefore, it has

$$\frac{1}{N}\sum_{i=1}^{N}\mathcal{L}\left(\mathcal{F}^*_{S'+\delta^w}(x'_i + \delta^w), y_i\right) \leq \frac{1}{N}\sum_{i=1}^{N}\mathcal{L}\left(G^*(x'_i + \delta^w), y_i\right) = a.$$

In fact, $\frac{1}{N}\sum_{i=1}^{N}\mathcal{L}\left(\mathcal{F}^*_{S'+\delta^w}(x'_i + \delta^w), y_i\right) = a$. This is because, if

$$\frac{1}{N}\sum_{i=1}^{N}\mathcal{L}\left(\mathcal{F}^*_{S'+\delta^w}(x'_i + \delta^w), y_i\right) = b < a,$$

let $H^* = \mathcal{F}^*_{S'+\delta^w}(t + \delta^w)$, it has

$$\frac{1}{N}\sum_{i=1}^{N}\mathcal{L}\left(H^*(x'_i), y_i\right) = \frac{1}{N}\sum_{i=1}^{N}\mathcal{L}\left(\mathcal{F}^*_{S'+\delta^w}(x'_i + \delta^w), y_i\right) = b < a,$$

violating the condition that $a = \min_{\mathcal{F}} \frac{1}{N}\sum_{i=1}^{N}\mathcal{L}\left(\mathcal{F}(x'_i), y_i\right)$.

Therefore, it has

$$\mathcal{F}^*_{S'+\delta^w}(t) = \mathcal{F}^*_{S'}(t - \delta^w).$$

Then we have

$$\frac{1}{N}\sum_{i=1}^{N}\mathcal{L}\left(\mathcal{F}^*_{S'+\delta^w}(x'_i), y_i\right) = \frac{1}{N}\sum_{i=1}^{N}\mathcal{L}\left(\mathcal{F}^*_{S'}(x'_i - \delta^w), y_i\right)$$

Now we use the McDiarmid's inequality to complete the proof of the first part. For each $(x_i, y_i)$, for different $\delta^w$ and $\bar{\delta}^w$ on one dimension, it holds that

$$\left|\mathcal{L}\left(\mathcal{F}^*_{S'}(x'_i - \delta^w), y_i\right) - \mathcal{L}\left(\mathcal{F}^*_{S'}(x'_i - \bar{\delta}^w), y_i\right)\right|$$
$$= \left|\log\left(1 + e^{y_i \cdot \mathcal{F}^*_{S'}(x'_i - \delta^w)}\right) - \log\left(1 + e^{y_i \cdot \mathcal{F}^*_{S'}(x'_i - \bar{\delta}^w)}\right)\right|$$
$$\leq \left|\mathcal{F}^*_{S'}(x'_i - \delta^w) - \mathcal{F}^*_{S'}(x'_i - \bar{\delta}^w)\right|$$
$$= \left|W^L \frac{1}{\sqrt{d_{L-1}}}\text{ReLU}\left(W^{L-1}\cdots\frac{1}{\sqrt{d_2}}\text{ReLU}\left(W^2\text{ReLU}\left(\frac{1}{\sqrt{d_1}}W^1(x_i + \delta^w) + b^1\right) + b^2\right) + \cdots + b^{L-1}\right) + b^L\right.$$
$$\left. - W^L \frac{1}{\sqrt{d_{L-1}}}\text{ReLU}\left(W^{L-1}\cdots\frac{1}{\sqrt{d_2}}\text{ReLU}\left(W^2\text{ReLU}\left(\frac{1}{\sqrt{d_1}}W^1(x_i + \bar{\delta}^w) + b^1\right) + b^2\right) + \cdots + b^{L-1}\right) + b^L\right|$$
$$\leq \frac{1}{\sqrt{d}}\frac{1}{\sqrt{d_{L-1}d_{L-2}\cdots d_2}}||W^L||_{1,\infty}\cdots||W^2||_{1,\infty}||W^1||_{1,\infty}\epsilon_w.$$

By McDiarmid's inequality, let $c = \frac{1}{\sqrt{d}}\frac{1}{\sqrt{d_{L-1}d_{L-2}\cdots d_2}}||W^L||_{1,\infty}\cdots||W^2||_{1,\infty}||W^1||_{1,\infty}\epsilon_w$, it holds that

$$\frac{1}{N}\sum_{i=1}^{N}\mathcal{L}\left(\mathcal{F}^*_{S'}(x'_i - \delta^w), y_i\right) \leq \mathbb{E}_\eta\frac{1}{N}\sum_{i=1}^{N}\mathcal{L}\left(\mathcal{F}^*_{S'}(x'_i - \eta), y_i\right) + \alpha$$

with probability at least $1 - \exp\left(-\frac{2\alpha^2}{q \cdot c^2}\right)$, where $\eta$ obey the distribution $\mathcal{U}\{-\epsilon_w, \epsilon_w\}^q$.

Due to the symmetry of $\eta$, it always has

$$\mathbb{E}_\eta \frac{1}{N} \sum_{i=1}^N \mathcal{L}\left(\mathcal{F}_{S'}^*(x_i' - \eta), y_i\right) = \mathbb{E}_\eta \frac{1}{N} \sum_{i=1}^N \mathcal{L}\left(\mathcal{F}_{S'}^*(x_i' + \eta), y_i\right).$$

Therefore, let $\omega = \exp\left(-\frac{2\alpha^2}{q \cdot c^2}\right)$, it holds that

$$\frac{1}{N} \sum_{i=1}^N \mathcal{L}\left(\mathcal{F}_{S'}^*(x_i' - \delta^w), y_i\right) \leq \mathbb{E}_\eta \frac{1}{N} \sum_{i=1}^N \mathcal{L}(\mathcal{F}_{S'}^*(x_i' + \eta), y_i) + \alpha$$

$$\leq \mathbb{E}_\eta \frac{1}{N} \sum_{i=1}^N \mathcal{L}\left(\mathcal{F}_{S'}^*(x_i' + \eta), y_i\right) + O\left(\epsilon_w \sqrt{\frac{q \log 1/\omega}{d}}\right)$$

Then we will prove that
$$\mathcal{R}(\mathcal{D}', \mathcal{F}_{S'+\delta^w}^*) = \mathbb{E}_{(x',y)\sim\mathcal{D}'} \mathcal{L}\left(\mathcal{F}_{S'+\delta^w}^*(x'), y\right)$$

$$\leq \frac{1}{N} \sum_{i=1}^N \mathcal{L}\left(\mathcal{F}_{S'+\delta^w}^*(x_i'), y_i\right) + O\left(\sqrt{\frac{\log d}{N}}\right) + O\left(\sqrt{\frac{L}{N}}\right) + O\left(\sqrt{\frac{\log 1/\omega}{N}}\right).$$

By [57], when loss function $L(\mathcal{F}(x), y)$ is bounded by $[0, B]$, with probability at least $1 - \omega$, it holds that

$$\mathcal{R}(\mathcal{D}', \mathcal{F}) \leq \frac{1}{N} \sum_{i=1}^N \mathcal{L}\left(\mathcal{F}(x_i'), y_i\right) + 2B \cdot \mathrm{Rad}_{(x_i', y_i)\in S'}\left(\mathcal{L}(\mathcal{F})\right) + 3B\sqrt{\frac{1}{2N} \log \frac{2}{\omega}}$$

The remaining issue is to compute $\mathrm{Rad}_{(x_i', y_i)\in S'}\left(\mathcal{L}(\mathcal{F}_{S'+\delta^w}^*)\right)$.

As loss function $L(z) = \log(1 + e^{-z})$ is 1-Lipschitz under $z = y \cdot \mathcal{F}_{S'+\delta^w}^*(x)$, by Talagrand's contraction lemma [49, 58],

$$\mathrm{Rad}_{(x_i', y_i)\in S'}\left(\mathcal{L}(\mathcal{F}_{S'+\delta^w}^*)\right) = \mathbb{E}_{\sigma_i\in\{-1,+1\}} \left[\sup_{L(\mathcal{F}_{S'+\delta^w}^*)} \frac{1}{N} \sum_{i=1}^N \sigma_i L(\mathcal{F}_{S'+\delta^w}^*(x_i'), y_i)\right]$$

$$\leq \mathbb{E}_{\sigma_i\in\{-1,+1\}} \left[\sup_{\mathcal{F}_{S'+\delta^w}^*} \frac{1}{N} \sum_{i=1}^N \sigma_i y_i \mathcal{F}_{S'+\delta^w}^*(x_i')\right]$$

$$= \mathbb{E}_{\sigma_i\in\{-1,+1\}} \left[\sup_{\mathcal{F}_{S'+\delta^w}^*} \frac{1}{N} \sum_{i=1}^N \sigma_i \mathcal{F}_{S'+\delta^w}^*(x_i')\right]$$

$$= \mathrm{Rad}_{(x_i', y_i)\in S'}\left(\mathcal{F}_{S'+\delta^w}^*\right)$$

From Theorem 1 in [84], it holds that

$$\mathrm{Rad}_{(x_i', y_i)\in S'}\left(\mathcal{F}_{S'+\delta^w}^*\right) \leq \prod_{l=1}^L \left(\|W^l\|_{1,\infty} + \|b^l\|_\infty\right) \left(\sqrt{\frac{(L+2)\log 4}{N}} + \sqrt{\frac{2\log(2d)}{N}}\right)$$

Therefore, it has

$$\mathcal{R}\left(\mathcal{D}', \mathcal{F}_{S'+\delta^w}^*\right) \leq \frac{1}{N} \sum_{i=1}^N \mathcal{L}\left(\mathcal{F}_{S'+\delta^w}^*(x_i'), y_i\right) + \log\left(1 + \exp\left(\max_x \mathcal{F}_{S'+\delta^w}^*(x)\right)\right) \cdot$$

$$\left(2 \cdot \mathrm{Rad}_{(x_i', y_i)\in S'}(\mathcal{L}(\mathcal{F}_{S'+\delta^w}^*)) + 3\sqrt{\frac{1}{2N} \log \frac{2}{\omega}}\right)$$

$$\leq \frac{1}{N} \sum_{i=1}^N \mathcal{L}\left(\mathcal{F}_{S'+\delta^w}^*(x_i' + \delta^w), y_i\right) + O\left(\sqrt{\frac{\log d}{N}}\right) + O\left(\sqrt{\frac{L}{N}}\right) + O\left(\sqrt{\frac{\log 1/\omega}{N}}\right)$$

$$+ O\left(\sqrt{\frac{q\epsilon_w \log 1/\omega}{d}}\right) + O\left(\frac{\epsilon_w}{\sqrt{d}}\right)$$

$\square$

**Theorem B.13** (Theorem 5.6, restated). *With probability at least $1 - \omega$ of the (unrestricted) poisoned dataset $\{(x_i + \delta_i^p, y_i)\}_{i=1}^N = S' \sim \mathcal{D}'$, it holds that*

$$\mathcal{R}\left(\mathcal{D}', \mathcal{F}_{S'|_{\mathcal{P}}}^*\right) \leq \frac{1}{N}\sum_{i=1}^N \mathcal{L}\left(\mathcal{F}_{S'|_{\mathcal{P}}}^*\left(x_i + \delta_i^p|_{\mathcal{P}}\right), y_i\right) + O\left(\frac{q\epsilon_p}{\sqrt{d}}\right)$$

$$+ O\left(\sqrt{\frac{\log d}{N}}\right) + O\left(\sqrt{\frac{L}{N}}\right) + O\left(\sqrt{\frac{\log 1/\omega}{N}}\right).$$

*Proof of Theorem 5.6.* For every $i$,

$$\mathcal{L}\left(\mathcal{F}_{S'|_{\mathcal{P}}}^*\left(x_i + \delta_i^p\right), y_i\right) - \mathcal{L}\left(\mathcal{F}_{S'|_{\mathcal{P}}}^*\left(x_i + \delta_i^p|_{\mathcal{P}}\right), y_i\right)$$

$$\leq \left|\mathcal{F}_{S'|_{\mathcal{P}}}^*\left(x_i + \delta_i^p\right) - \mathcal{F}_{S'|_{\mathcal{P}}}^*\left(x_i + \delta_i^p|_{\mathcal{P}}\right)\right|$$

$$= \left|W^L \frac{1}{\sqrt{d_{L-1}}}\text{ReLU}\left(W^{L-1}\cdots\frac{1}{\sqrt{d_2}}\text{ReLU}\left(L_2\right) + \cdots + b^{L-1}\right) + b^L\right.$$

$$\left. - W^L \frac{1}{\sqrt{d_{L-1}}}\text{ReLU}\left(W^{L-1}\cdots\frac{1}{\sqrt{d_2}}\text{ReLU}\left(L_2\right) + \cdots + b^{L-1}\right) + b^L\right|$$

$$\leq \frac{1}{\sqrt{d}}\frac{1}{\sqrt{d_{L-1}d_{L-2}\cdots d_2}}\|W^L\|_{1,\infty}\|W^{L-1}\|_{1,\infty}\cdots\|W^2\|_{1,\infty}\|W^1\delta_i^p|_{[d]-\mathcal{P}}\|_1$$

$$\leq \frac{1}{\sqrt{d_{L-1}d_{L-2}\cdots d_2}}\|W^L\|_{1,\infty}\|W^{L-1}\|_{1,\infty}\cdots\|W^2\|_{1,\infty}\|W^1\|_{1,\infty} \cdot \frac{|[d] - \mathcal{P}| \cdot \|\delta_i^p\|_\infty}{\sqrt{d}}$$

$$\leq \frac{1}{\sqrt{d_{L-1}d_{L-2}\cdots d_2}}\|W^L\|_{1,\infty}\|W^{L-1}\|_{1,\infty}\cdots\|W^2\|_{1,\infty}\|W_1\|_{1,\infty} \cdot \frac{q\epsilon_p}{\sqrt{d}}.$$

where $L_2 = W^2\text{ReLU}\left(\frac{1}{\sqrt{d_1}}W^1\left(x_i + \delta_i^p\right) + b^1\right) + b^2$.

Therefore, it holds that

$$\frac{1}{N}\sum_{i=1}^N \mathcal{L}\left(\mathcal{F}_{S'|_{\mathcal{P}}}^*(x_i + \delta_i^p), y_i\right) \leq \frac{1}{N}\sum_{i=1}^N \mathcal{L}\left(\mathcal{F}_{S'|_{\mathcal{P}}}^*(x_i + \delta_i^p|_{\mathcal{P}}), y_i\right) + O\left(\frac{q\epsilon_p}{\sqrt{d}}\right).$$

Then, similar to the proof of Theorem 5.2, it has

$$\mathcal{R}(\mathcal{D}', \mathcal{F}_{S'|_{\mathcal{P}}}^*) \leq \frac{1}{N}\sum_{i=1}^N \mathcal{L}\left(\mathcal{F}_{S'|_{\mathcal{P}}}^*(x_i + \delta_i^p), y_i\right) + \log\left(1 + \exp\left(\max_x \mathcal{F}_{S'|_{\mathcal{P}}}^*(x)\right)\right) \cdot$$

$$\left(2 \cdot \text{Rad}_{(x_i', y_i)\in S'}\left(\mathcal{L}(\mathcal{F}_{S'|_{\mathcal{P}}}^*)\right) + 3\sqrt{\frac{1}{2N}\log\frac{2}{\omega}}\right)$$

$$\leq \frac{1}{N}\sum_{i=1}^N \mathcal{L}\left(\mathcal{F}_{S'|_{\mathcal{P}}}^*(x_i + \delta_i^p), y_i\right) + O\left(\sqrt{\frac{\log d}{N}}\right) + O\left(\sqrt{\frac{L}{N}}\right) + O\left(\sqrt{\frac{\log 1/\omega}{N}}\right)$$

$$\leq \frac{1}{N}\sum_{i=1}^N \mathcal{L}\left(\mathcal{F}_{S'|_{\mathcal{P}}}^*(x_i + \delta_i^p|_{\mathcal{P}}), y_i\right) + O\left(\frac{q\epsilon_p}{\sqrt{d}}\right) + O\left(\sqrt{\frac{\log d}{N}}\right)$$

$$+ O\left(\sqrt{\frac{L}{N}}\right) + O\left(\sqrt{\frac{\log 1/\omega}{N}}\right).$$

$\square$

**Corollary B.14** (Corollary 5.7, restated). *With probability at least $1 - 3\omega$ for the restricted poisoned dataset $S'|_{\mathcal{P}} \sim \mathcal{D}'|_{\mathcal{P}}$ and the key $\zeta \in \mathbb{R}^d$ selected from certain distribution, we can craft the watermark $\delta^w$ satisfied:*

$$\mathcal{R}(\mathcal{D}', \mathcal{F}^*_{\tilde{S}}) \leq \mathbb{E}_\eta \frac{1}{N} \sum_{i=1}^N \mathcal{L}\left(\mathcal{F}^*_{S|_{\mathcal{P}}}(x_i + \delta^p_i|_{\mathcal{P}} + \eta), y_i\right)$$

$$+ O\left(\sqrt{\frac{L}{N}}\right) + O\left(\sqrt{\frac{\log d}{N}}\right) + O\left(\sqrt{\frac{\log 1/\omega}{N}}\right) + O\left(\frac{q\epsilon_p}{\sqrt{d}}\right) + O\left(\epsilon_w\sqrt{\frac{q\log 1/\omega}{d}}\right),$$

*where $\tilde{S} = S|_{\mathcal{P}} + \delta^w$ is the watermarked dataset.*

*Proof.* By Theorem 5.2, with probability at least $1 - 2\omega$ it holds that

$$\mathcal{R}(\mathcal{D}', \mathcal{F}^*_{\tilde{S}}) \leq \mathbb{E}_\eta \frac{1}{N} \sum_{i=1}^N \mathcal{L}\left(\mathcal{F}^*_{S|_{\mathcal{P}}}(x_i + \delta^i_p + \eta), y_i\right) + O\left(\sqrt{\frac{L}{N}}\right)$$

$$+ O\left(\sqrt{\frac{\log d}{N}}\right) + O\left(\sqrt{\frac{\log 1/\omega}{N}}\right) + O\left(\epsilon_w\sqrt{\frac{q\log 1/\omega}{d}}\right).$$

By Theorem 5.6, with probability at least $1 - \omega$, for every $\eta$, it holds that

$$\frac{1}{N} \sum_{i=1}^N \mathcal{L}\left(\mathcal{F}^*_{S'|_{\mathcal{P}}}(x_i + \delta^p_i + \eta), y_i\right) \leq \frac{1}{N} \sum_{i=1}^N \mathcal{L}\left(\mathcal{F}^*_{S'|_{\mathcal{P}}}(x_i + \delta^p_i|_{\mathcal{P}} + \eta), y_i\right) + O\left(\frac{q\epsilon_p}{\sqrt{d}}\right).$$

Combine the above two inequalities directly to complete the proof. $\qquad\square$

# C   Watermarking Algorithm

---
**Algorithm 1** Post-Poisoning Watermarking
---
**Input:** The poisoned training dataset $D_{\mathcal{P}} = \{(x_i + \delta^p_i, y_i)\}_{i=1}^N$. The key $\zeta$.
**Output:** Watermarked training dataset $D_{\mathcal{W}} = \{(x_i + \delta^p_i + \delta^w, y_i)\}_{i=1}^N$.
Choose the watermarking dimension $\mathcal{W}$.
Set $\delta^w = \epsilon_w \cdot \text{sign}(\zeta)|_{\mathcal{W}}$.

---

---
**Algorithm 2** Poisoning-Concurrent Watermarking
---
**Input:** The training dataset $D_{\mathcal{P}} = \{(x_i, y_i)\}_{i=1}^N$. The key $\zeta$.
**Output:** Watermarked poisoned training dataset $D_{\mathcal{W}} = \{(x_i + \delta^p_i + \delta^w, y_i)\}_{i=1}^N$.
Choose the watermarking dimension $\mathcal{W}$.
Set $\delta^w = \epsilon_w \cdot \text{sign}(\zeta)|_{\mathcal{W}}$.
Update poisons $\delta^p_i$ on poisoning dimension $\mathcal{P} = [d] - \mathcal{W}$.

---

---
**Algorithm 3** Detection
---
**Input:** The suspect training data $\tilde{x}$. The key $\zeta$. The detection threshold $\tau$.
**Output:** 1 (Positive) or 0 (Negative).
Compute the detection value $v = \zeta^T \tilde{x}$
If $v > \tau$, return 1, else $v \leq \tau$, return 0.

---

# D   Additional Experiments

## D.1   Additional Experiments on More Datasets

We extend our evaluation to CIFAR-100 and TinyImageNet for UE and AP poisons on Table 3 and Table 4 respectively. Results demonstrate similar trends: as the watermark length increases,

Table 3: The clean accuracy (Acc, %) and AUROC of UE and AP availability attacks both on post-poisoning watermarking and poisoning-concurrent watermarking with different watermarking length $q$ under ResNet-18 and CIFAR-100.

| Length/Method | UE | | AP | |
|---|---|---|---|---|
| Acc($\downarrow$)/AUROC($\uparrow$) | Post-Poisoning | Poisoning-Concurrent | Post-Poisoning | Poisoning-Concurrent |
| 0(Baseline) | 1.24/- | 1.24/- | 1.71/- | 1.71/- |
| 100 | 1.21/0.5796 | 1.15/0.8064 | 1.75/0.5913 | 1.66/0.6950 |
| 300 | 1.25/0.7145 | 1.43/0.8839 | 1.66/0.7667 | 1.69/0.7732 |
| 500 | 1.19/0.7822 | 2.51/0.9150 | 1.77/0.8710 | 1.85/0.8931 |
| 1000 | 1.43/0.9354 | 1.46/0.9758 | 1.72/0.9669 | 2.36/0.9949 |
| 1500 | 1.10/0.9963 | 1.66/0.9992 | 1.68/0.9893 | 2.19/0.9995 |
| 2000 | 1.28/0.9982 | 3.49/0.9999 | 1.90/0.9986 | 6.98/1.0000 |
| 2500 | 1.36/0.9995 | 54.10/1.0000 | 1.76/1.0000 | 32.41/1.0000 |
| 3000 | 1.57/1.0000 | 71.04/1.0000 | 2.36/1.0000 | 69.85/1.0000 |

Table 4: The clean accuracy (Acc, %) and AUROC of UE and AP availability attacks both on post-poisoning watermarking and poisoning-concurrent watermarking with different watermarking length $q$ under ResNet-18 and TinyImageNet.

| Length/Method | UE | | AP | |
|---|---|---|---|---|
| Acc($\downarrow$)/AUROC($\uparrow$) | Post-Poisoning | Poisoning-Concurrent | Post-Poisoning | Poisoning-Concurrent |
| 0(Baseline) | 0.75/- | 0.75/- | 9.37/- | 9.37/- |
| 500 | 1.15/0.7850 | 1.24/0.9623 | 11.85/0.8054 | 8.74/0.9794 |
| 1000 | 0.92/0.8587 | 1.70/0.9952 | 8.62/0.8620 | 11.25/0.9967 |
| 2000 | 0.95/0.9596 | 3.69/0.9994 | 13.40/0.9640 | 22.61/0.9998 |
| 5000 | 2.23/0.9998 | 11.01/0.9999 | 22.17/1.0000 | 43.30/1.0000 |
| 10000 | 7.14/1.0000 | 48.32/1.0000 | 36.81/1.0000 | 47.05/1.0000 |

detectability improves (higher AUROC), while poisoning effectiveness decreases (higher clean accuracy), confirming our theoretical claims.

Furthermore, for text dataset, we implement watermarking ($\epsilon_w = 16/255$ ) in a backdoor attack on SST-2 dataset with BERT-base model [24], observing similar trends compared with other visual datasets for this NLP task.

Table 5: The accuracy (Acc, %), ASR and AUROC of SST-2 dataset on BERT-base model with different watermarking length $q$.

| | Post-Poisoning | Poisoning-Concurrent |
|---|---|---|
| Length | Acc/ASR/AUROC | Acc/ASR/AUROC |
| 0 | 89.7/98.0/- | 89.7/98.0/- |
| 100 | 89.8/97.8/0.697 | 89.6/97.2/0.969 |
| 200 | 89.2/97.3/0.852 | 89.9/96.1/0.983 |
| 400 | 89.6/96.2/0.931 | 89.3/90.5/0.998 |
| 600 | 89.3/96.7/0.983 | 89.5/72.3/0.999 |

## D.2   Additional Experiments on More Network Structures

For model transferability, we evaluate our watermarking with length $q = 1000$ across ResNet-50, VGG-19, DenseNet121, WRN34-10, MobileNet v2 and ViT-B models. Results shown in Table 6 and Table 7 demonstrate strong transferability (high AUROC and low accuracy) across network architectures, further validating our theoretical insights.

## D.3   Results under Different Watermarking Budget

We evaluate our watermarking algorithms under different watermarking budgets—$4/255$, $8/255$, $16/255$, and $32/255$—with a fixed watermarking length of 1000. The results indicate that as the budget $\epsilon_w$ increases, detectability improves while poisoning effectiveness declines. This aligns with

Table 6: The clean accuracy (Acc, %), attack success rate (ASR, %), and AUROC of Narcissus and AdvSc backdoor attacks on both post-poisoning watermarking and poisoning-concurrent watermarking with various victim models under CIFAR-10.

| Model/Method | Narcissus | | AdvSc | |
| Acc/ASR/AUROC($\uparrow$) | Post-Poisoning | Poisoning-Concurrent | Post-Poisoning | Poisoning-Concurrent |
| --- | --- | --- | --- | --- |
| ResNet-18 | 94.40/92.43/0.9974 | 94.32/92.03/0.9992 | 93.05/94.41/0.9809 | 93.38/84.39/0.9995 |
| ResNet-50 | 94.46/93.12/0.9969 | 94.85/93.01/0.9985 | 92.55/93.30/0.9827 | 92.16/86.53/0.9995 |
| VGG-19 | 93.74/91.80/0.9975 | 92.61/91.97/0.9995 | 91.47/93.94/0.9926 | 91.80/79.34/0.9999 |
| DenseNet121 | 94.18/92.66/0.9977 | 94.52/92.39/0.9990 | 94.12/93.73/0.9905 | 92.67/90.32/0.9998 |
| WRN34-10 | 94.95/92.14/0.9981 | 95.02/91.36/0.9989 | 94.74/94.85/0.9860 | 94.12/89.63/0.9994 |
| MobileNet v2 | 94.63/92.41/0.9972 | 94.15/92.14/0.9986 | 93.63/94.51/0.9754 | 93.75/83.29/0.9996 |
| ViT-B | 94.87/94.25/0.9991 | 95.25/93.37/1.0000 | 94.32/93.26/0.9922 | 94.23/91.45/1.000 |

Table 7: The clean accuracy (Acc,%) and AUROC of UE and AP availability attacks both on post-poisoning watermarking and poisoning-concurrent watermarking with various victim models under CIFAR-10.

| Model/Method | UE | | AP | |
| Acc($\downarrow$)/AUROC($\uparrow$) | Post-Poisoning | Poisoning-Concurrent | Post-Poisoning | Poisoning-Concurrent |
| --- | --- | --- | --- | --- |
| ResNet-18 | 11.37/0.9499 | 9.42/0.9991 | 10.58/0.9742 | 21.87/0.9949 |
| ResNet-50 | 10.15/0.9583 | 12.26/0.9992 | 9.97/0.9678 | 14.76/0.9947 |
| VGG-19 | 12.96/0.9644 | 12.21/0.9993 | 10.80/0.9800 | 20.34/0.9952 |
| DenseNet121 | 19.30/0.9545 | 17.87/0.9985 | 12.35/0.9767 | 11.76/0.9978 |
| WRN34-10 | 12.31/0.9702 | 10.55/0.9988 | 10.24/0.9821 | 15.98/0.9958 |
| MobileNet v2 | 14.03/0.9473 | 16.90/0.9986 | 11.36/0.9726 | 18.51/0.9941 |
| ViT-B | 13.97/0.9728 | 14.80/0.9989 | 10.51/0.9793 | 12.75/0.9970 |

our theoretical findings: as $\epsilon_w$ grows, both $\Omega(\sqrt{d}/\epsilon_w)$ (post-poisoning) and $\Omega(1/\epsilon_w^2)$ (poisoning-concurrent) decrease, leading to better detectability. Additionally, the error term $O\left(\epsilon_w \sqrt{\frac{q \log 1/\omega}{d}}\right)$ (Theorem 5.2 and Corollary 5.7) influences poisoning effectiveness, meaning a larger $\epsilon_w$ weakens the poisoning power guarantee. This is evident in our results, where AdvSc achieves only 60.04% and 36.45% ASR under $\epsilon_w = 32/255$ for post-poisoning and poisoning-concurrent watermarking, a trend also observed in Figure 1 in Section 6.3.

Table 8: The clean accuracy (Acc, %), attack success rate (ASR, %), and AUROC of Narcissus and AdvSc backdoor attacks on both post-poisoning watermarking and poisoning-concurrent watermarking under different watermarking budgets on CIFAR-10 dataset.

| Budget/Method | Narcissus | | AdvSc | |
| Acc/ASR/AUROC($\uparrow$) | Post-Poisoning | Poisoning-Concurrent | Post-Poisoning | Poisoning-Concurrent |
| --- | --- | --- | --- | --- |
| 4/255 | 94.35/94.28/0.9114 | 94.43/94.21/0.8297 | 92.94/98.68/0.8132 | 93.25/91.68/0.8655 |
| 8/255 | 94.71/93.76/0.9535 | 94.99/92.69/0.8948 | 93.04/98.88/0.9427 | 93.27/87.48/0.9651 |
| 16/255 | 94.40/92.43/0.9974 | 94.32/92.03/0.9992 | 93.05/94.41/0.9809 | 93.38/84.39/0.9995 |
| 32/255 | 94.86/90.66/0.9998 | 94.87/80.17/1.0000 | 93.13/60.04/0.9999 | 92.76/36.45/1.0000 |

## D.4 Watermarking on Clean Samples

Beyond data poisoning, we test watermarking on clean CIFAR-10 with $\epsilon_w$ be 4/255, 8/255, 16/255 and 32/255 on Table 9. The results indicate strong detectability with minimal accuracy degradation, even for large perturbations (32/255). It is worth noting that, for clean samples, post-poisoning and poisoning-concurrent watermarking will become the same as there are no poisons involved.

## D.5 Computational Cost

We evaluate the computational overheads for our watermarking techniques on UE and AP availability attacks, as well as Narcissus and AdvSc backdoor attacks. All experiments are evaluated on a single NVIDIA A800 80GB PCIe GPU. Results in Table 10 show that our watermarking is highly efficient,

Table 9: The accuracy (Acc, %) and AUROC of clean CIFAR-10 dataset with different watermarking length $q$ under ResNet-18.

| Budget Length | 4/255 Acc/AUROC | 8/255 Acc/AUROC | 16/255 Acc/AUROC | 32/255 Acc/AUROC |
|---|---|---|---|---|
| 0 | 95.25/- | 95.25/- | 95.25/- | 95.25/- |
| 200 | 95.12/0.5527 | 94.85/0.6218 | 94.75/0.7854 | 94.48/0.8672 |
| 500 | 94.90/0.6638 | 94.53/0.8317 | 93.66/0.9683 | 91.66/0.9990 |
| 1000 | 94.56/0.8679 | 94.08/0.9700 | 92.87/0.9929 | 89.54/1.0000 |
| 1500 | 94.22/0.9491 | 93.82/0.9764 | 92.02/0.9998 | 91.60/1.0000 |
| 2000 | 94.01/0.9736 | 93.37/0.9946 | 90.34/1.0000 | 88.20/1.0000 |
| 2500 | 93.86/0.9935 | 93.49/1.0000 | 88.70/1.0000 | 83.20/1.0000 |

requiring only seconds for post-poisoning watermarking and detection. Even for poisoning-concurrent watermarking, it incurs a minimal 10-minute overhead. Therefore, we believe our watermarking schemes are efficient to deploy in real-world applications.

Table 10: The time cost of our watermarking techniques under CIFAR-10 dataset on various data poisoning attacks.

| Time | UE | AP | Narcissus | AdvSc |
|---|---|---|---|---|
| Poisoning(baseline) | ≈80min | ≈65min | ≈70min | ≈190min |
| Post-poisoning | ≈30s | ≈30s | ≈30s | ≈30s |
| Poisoning-concurrent | ≈90min | ≈70min | ≈75min | ≈200min |
| Detection | ≈40s | ≈40s | ≈40s | ≈40s |

## E  Robust Watermarking under Various Defenses and Removals

**Data augmentation and image regeneration.** Under some data augmentations or image reconstructions, the provable watermarking may not hold because the relative position between watermarks and keys has been broken. However, we can train a watermark detector with the known key, and judge whether the data is watermarked with the detector. Specifically, denote the clean dataset as $\{(x_i, y_i)\}_{i=1}^N$, where $x_i \in \mathbb{R}^d$ is the data, $y_i \in \mathbb{Z}$ is the label. The key $\zeta \in \mathbb{R}^d$. We craft the watermark detection training set $\mathcal{D}_d$ as $\{(x_i, 0)\}_{i=1}^N \cap \{(x_i + \epsilon_w \cdot \zeta, 1)\}_{i=1}^N$, where $\epsilon_w$ is a small budget that the injected watermarks $\delta^w$ compromise, and train a detector $\mathcal{T}$ with $\mathcal{D}_d$ under data augmentations. For a suspect data $\tilde{x}$ which may be poisoned with watermarking, we argue that $\tilde{x}$ is poisoned if $\mathcal{T}(\tilde{x}) = 1$; otherwise, $\tilde{x}$ is recognized as benign data. We evaluate the performance of our detector under several data augmentations, including Random Flip, Cutout [16], Color Jitter and Grayscale. Furthermore, we also evaluate the watermarking performance under some regeneration attacks including VAE-based attack [14] and generative adversarial network [78]. Experimental results presented in Table 11 have shown stronger detection performance, validating the robustness of our proposed watermarking.

Table 11: The detection performance (AUROC) of poisoning-concurrent watermarking of UE and AP with watermarking length be 500 under various data augmentations.

| Type | Random Flip | Cutout | Color Jitter | Grayscale | VAE [14] | GAN [78] |
|---|---|---|---|---|---|---|
| UE | 1.0000 | 1.0000 | 1.0000 | 0.9930 | 0.9987 | 0.9853 |
| AP | 1.0000 | 1.0000 | 1.0000 | 0.9996 | 0.9395 | 0.9830 |

**Differential privacy noises.** To further evaluate the robustness of our watermarking, we consider adaptive attacks based on $(\epsilon, \delta)$-DP, applying both Gaussian and Laplacian mechanisms with $\epsilon = 2, \delta = 10^{-5}$. We evaluate them on poisoning-concurrent watermarking with $q = 1500$ under UE, AP, Narcissus and AdvSc, results are shown in Table 12. Unfortunately, due to the extremely large noise level introduced in the pixel space (e.g., $\sigma = \frac{\Delta}{\epsilon} = \frac{8/255 \cdot 3072}{2} = 48$, for the Laplacian mechanism) to

the pixel space, the network fails to converge. This is because DP mechanisms are typically applied to neural network gradients or parameters, not directly to training data, and the severe perturbation causes samples from different classes to become indistinguishable.

It may be counterintuitive that UE and AP achieve lower clean accuracy under DP noise. Under normal training, UE and AP can still converge, reaching nearly 100% training and validation accuracy but only about 10% test accuracy, consistent with availability attack objectives. In contrast, when training on DP-perturbed data, the training/validation accuracy also drops to about 10%, indicating complete training failure. This contradicts the goal of availability attacks, which aim to deceive victims into believing the model is well-trained, while failing on unseen test data (see [37] for details). Notably, backdoor attacks don't exhibit this confusion as they seek high ASR rather than low accuracy. Although DP-based defenses reduce the detection performance of watermarking, the poisoning utilities have been completely destroyed. Therefore, DP-based defenses are not applicable in our context.

Table 12: Clean accuracy(Acc, %), attack success rate(ASR, %) and AUROC of poisoning-concurrent watermarking with length be 1500 under DP noises.

| ACC/ASR/AUROC | DP-Gaussian | DP-Laplacian |
|---|---|---|
| UE | 14.01/-/0.8016 | 12.79/-/0.5759 |
| AP | 15.85/-/0.7923 | 10.88/-/0.6232 |
| Narcissus | 13.37/10.12/0.8135 | 11.76/9.98/0.6126 |
| AdvSc | 15.11/10.03/0.7447 | 11.15/10.06/0.5880 |

**Diffusion purification.** For diffusion purification [94], results are shown in Table 13. Although our watermarking exhibits weak detectability, it is important to note that the poison utility is simultaneously eliminated. As shown in the following table, diffusion purification significantly mitigates availability poisoning attacks, recovering test accuracy from about 10% to over 80%. It also destroys backdoor poisoning attacks, reducing the attack success rate to less than 20%. This is reasonable as diffusion purification is a powerful defense against noise injection, including adversarial attacks [60], availability attacks [17] and diffusion model watermarking [34].

In our scenario, watermarking is designed to serve the purpose of data poisoning. If the poisoning itself is neutralized, the effectiveness of the watermark becomes irrelevant. Given that our work focuses on imperceptible poisoning and watermarking, this limitation appears to be an inherent trade-off. Similar to DP-based defenses, although diffusion purification reduces the detection performance of watermarking, the poisoning utilities have been completely destroyed. Therefore,diffusion purification is also not applicable in our context.

Table 13: Accuracy(ACC), attack success rate(ASR) and AUROC of poisoning-concurrent watermarking with length be 1500 under diffusion purification.

| Type | UE | AP | Narcissus | AdvSc |
|---|---|---|---|---|
| ACC/ASR/AUROC | 84.67/-/0.5251 | 85.22/-/0.5189 | 93.17/16.86/0.5375 | 93.08/10.01/0.5420 |

**Potential removal methods.** We conduct additional experiments on UE and AP with direct masking of the known watermarking dimensions (Masking), as well as the adversarial noising proposed by [54]. We test both post-poisoning and poisoning-concurrent watermarking under $q = 2000$.As the results shown below, although the detection performance (AUROC) drops, the utility of UE and AP also degrades significantly. The underlying reasons may be that availability attacks are designed with potential linear shortcut features [86, 98], the masking of watermarking dimensions somehow destroys these linear features, undermining the unlearnability (low Acc). Adversarial Noising further destroys the poisoning utility as availability attacks are theoretically removed by perfect adversarial training [76]. Therefore, these adaptive removal attacks fail to maintain the poisoning utility, making them not applicable in our cases.

Table 14: Accuracy(Acc) and AUROC of UE and AP availability attacks under potential removal methods, masking and adversarial noising.

| Acc/AUROC | Baseline | Masking | Adversarial Noising |
|---|---|---|---|
| UE(Post-Poisoning) | 9.06/0.9992 | 60.71/0.4998 | 72.90/0.5893 |
| AP(Post-Poisoning) | 10.48/0.9987 | 56.85/0.5005 | 76.21/0.5616 |
| UE(Poisoning-Concurrent) | 10.03/1.0000 | 55.49/0.5014 | 68.37/0.6206 |
| AP(Poisoning-Concurrent) | 38.62/1.0000 | 59.87/0.5002 | 74.63/0.5833 |

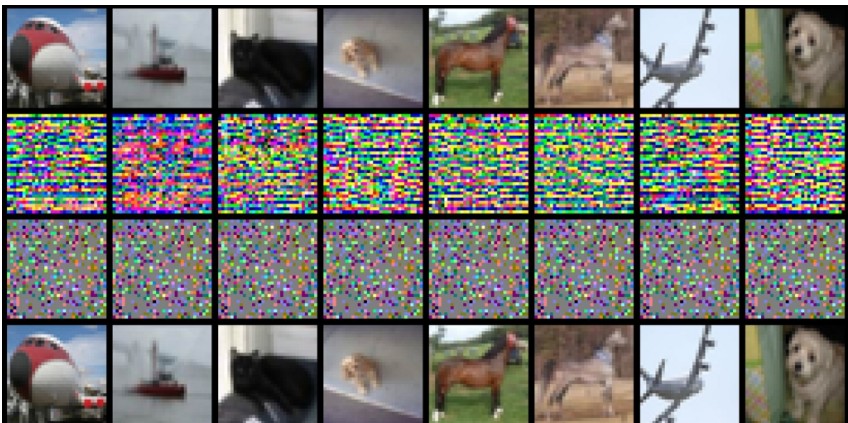

Figure 3: Visualization of UE poisoning-concurrent watermarking with length $q = 500$ for CIFAR-10 dataset. The first row is the benign images, the second row is the normalized UE poisons, the third row is the normalized watermarks, the fourth row is the perturbed images under watermarking poisons.

## F Visualization

To further substantiate the imperceptibility of our proposed watermarking, we visualize the benign images, poisons, watermarks, and modified images. Both poisons and watermarks are normalized to $[0, 1]$ in order to improve their visibility. Figure 3 shows the watermarking visualization under UE poisons; our watermarking demonstrates strong imperceptibility.

## G Covertness of Watermarking

For an practical watermarking, beyond their detectability, it also requires *covertness*. That means, if users do not obtain the watermarking key $\zeta$, it is hard for them to discern poisoned data and benign data. In other words, if the key $\zeta$ is random (independent from the watermarks $\delta^w$), the performance between poisoned data $x' + \delta^w$ and benign data $x$ under random key $\zeta$ will have negligible difference. We will prove this property for post-poisoning watermarking; the property of poisoning-concurrent watermarking also holds similarly.

**Theorem G.1** (Covertness for post-poisoning watermarking)**.** *For post-poisoning watermarking with watermarks $\delta^w$, assume that the poisoned data $x' = x + \delta_x^p$, and the benign data $\bar{x}$ are independently sampled from the data distribution $\mathcal{D}$. For the random identical key $\zeta \in \mathbb{R}^d$, it holds that $\mathbb{E}_\zeta \left[ \zeta^T (x' + \delta^w) \right] = \mathbb{E}_\zeta \left[ \zeta^T \tilde{x} \right]$. Furthermore, it holds that $\mathbb{P}_\zeta \left[ \left| \zeta^T (x' + \delta^w) - \zeta^T \tilde{x} \right| \leq \sqrt{\frac{d}{2} \log \frac{2}{\omega}} \right] > 1 - \omega$.*

*Proof of Theorem G.1.* As $\zeta \in \mathbb{R}^d$ is the random identical key, it holds that

$$\mathbb{E}_\zeta \left[ \zeta^T (x' + \delta^w) \right] = 0$$

as well as

$$\mathbb{E}_\zeta \left[ \zeta^T \tilde{x} \right] = 0.$$

Therefore, it has

$$\mathbb{E}_\zeta \left[ \zeta^T(x' + \delta^w) \right] = \mathbb{E}_\zeta \left[ \zeta^T \tilde{x} \right].$$

Additionally, as $x' + \delta^w$ and $\bar{x}$ both lie in $[0, 1]$, it always has

$$|\zeta^i(x' + \delta^w)^i - \zeta^i \tilde{x}^i| \leq |\zeta^i| = 1$$

for all $i, \zeta$.

Therefore, by McDiarmid's inequality, for any $\alpha > 0$, it has

$$\mathbb{P}_\zeta[|\zeta^T(x' + \delta^w - \tilde{x})| \geq \alpha] \leq 2e^{-\frac{2\alpha^2}{d}}.$$

Therefore, let

$$\omega = 2e^{-\frac{2\alpha^2}{d}},$$

it has

$$\alpha = \sqrt{\frac{d}{2} \log \frac{2}{\omega}}.$$

$\square$

*Remark* G.2. For post-poisoning watermarking, if a detector does not obtain the key, the expected predictions for the (watermarked) poisoned data and (unwatermarked) benign data are equal. Therefore, it is hard to detect watermarks without the key.

We validate this property on two backdoor attacks, Narcissus and AdvSc, and two availability attacks, UE and AP. We consider the post-poisoning watermarking with watermarking length $q = 2000$, and test the detection performance of the corresponding watermarking key and the random identical key independently from the watermarking $\delta^w$. The results shown in Figure 4 demonstrate that, if the detector just uses a random key for detection, the AUROC is approaching 0.5, meaning that it is ineffective and almost like a random guess.

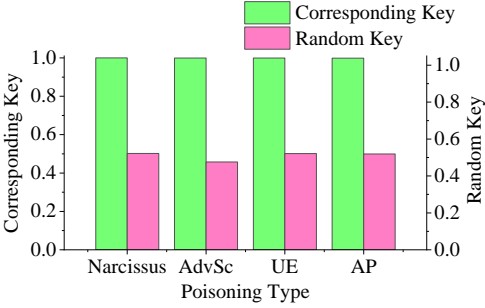

Figure 4: The detection performance (AUROC) of post-poisoning watermarking of several data poisoning attacks under corresponding key and a random key.

## H    Boarder Impact Statement

This paper aims at crafting watermarks for data poisoning attacks. As a method to ensure authorized users can identify potential data poisoning, we believe our work is beneficial to the community and does not have a negative social impact.

