# OpenReview forum: "Provable Watermarking for Data Poisoning Attacks"
_NeurIPS.cc/2025/Conference — NeurIPS 2025 poster_

### Official Review · Reviewer_aXke · 2025-06-21

**Clarity:** 2
**Significance:** 2
**Originality:** 3
**Rating:** 4
**Confidence:** 3

**Summary:**

The paper proposes a provably detectable watermarking schemes for poisoned datasets, aiming to bring transparency and accountability to the increasingly common use of data poisoning for protective purposes (e.g., copyright enforcement or model misuse prevention). The authors introduce two methods—post-poisoning watermarking and poisoning-concurrent watermarking—which embed subtle, random perturbations into poisoned data using a secret key. They provide theoretical guarantees on detection reliability, showing that the watermark can be statistically verified with high probability without harming the effectiveness of the original poisoning attack. The watermark is invisible to adversaries without the key and remains undetectable unless actively searched for using a tailored test. Experiments on CIFAR-10 across multiple poisoning strategies (e.g., backdoor and availability attacks) confirm that watermark detection is accurate and that the watermark preserves the utility of the poisoning. This work bridges the gap between data poisoning and verifiable data provenance by enabling poisoned data to be both effective and transparently marked.

**Questions:**

Can you clarify the practical scenarios in which your watermarking scheme would be deployed, and who the intended users are (e.g., researchers, data providers, regulators)? It would be helpful to understand a concrete use case where verifiable watermarking of poisoned data is both necessary and realistically actionable.

**Ethical Concerns:**

["NO or VERY MINOR ethics concerns only"]

**Final Justification:**

After carefully reading the rebuttal, I believe the authors have addressed most of my initial concerns. I provide the following justification for my updated recommendation:

Use Case Clarification: The authors provided a clear and detailed deployment scenario involving real-world actors, which significantly improved my understanding of the practical use of their watermarking scheme. While I still think the connection to real-world applications could be made more prominent earlier in the paper (e.g., through a motivating use case in the introduction), the rebuttal resolves my main concern regarding applicability.

Writing and Presentation: The authors acknowledged feedback regarding notations and presentation clarity and committed to improving these aspects by adding a symbol table and expanding experimental setup descriptions.

Experimental Scope: The authors expanded their experimental evaluation in the rebuttal, including new tasks (e.g., NLP with SST-2, multi modal with MS-COCO).

**Limitations:**

Yes

**Quality:**

2

**Strengths And Weaknesses:**

Strengths:
1. The paper beyond the empricall experiements and want to provide some theorical ganrantee

Weakness:
1. The overall setting feels disconnected from practical use cases. While the paper introduces a theoretically elegant watermarking mechanism for poisoned datasets, it's unclear who the target users of such a system would be in real-world scenarios. The authors mention legitimate uses of poisoning, such as for copyright protection or to prevent unauthorized model training, but these applications remain underdeveloped in the paper. There is little discussion of actual deployment contexts, potential adopters, or how the watermarking would interact with existing machine learning pipelines. Without a compelling, concrete example of where this would be applied in practice the proposed solution feels somewhat speculative.

2. The writing and presentation could be improved for clarity and accessibility. A notable issue is the inclusion of symbols like “d, ϵ” directly in the abstract and introduction without clear explanation. Readers will not understand what these symbols refer to, and dropping a theoretical scaling law into the abstract hurts readability. Throughout the paper, several mathematical notations and definitions are introduced abruptly, assuming too much background from the reader. Additionally, some sections, such as the experimental setup, could benefit from more structured exposition, as they currently mix theoretical and empirical points in a way that’s hard to follow. Improving clarity in writing would make the paper more accessible to a broader audience.

3. The scope of the experimental evaluation is too limited, especially given the paper’s broader claims. In the introduction, the authors highlight the importance of LLMs and large-scale model training, yet all experiments are conducted on small image datasets like CIFAR-10 using convolutional networks. This creates a significant mismatch between the motivation and the actual evaluation. The effectiveness of the proposed watermarking techniques for high-dimensional or complex data modalities (e.g., text or multimodal data used in LLMs) remains entirely untested. Moreover, real-world model training typically involves significant data preprocessing, augmentation, and noise, none of which are accounted for in the current experiments. As a result, it’s unclear how robust or applicable the method would be in more realistic and challenging settings.

---

> ### Author Rebuttal · Authors · 2025-07-31
>
> Thanks for your insightful comments and valuable suggestions. We provide responses below to address your concerns.
>
> **1. The overall setting feels disconnected from practical use cases. Can you clarify the practical scenarios in which your watermarking scheme would be deployed, and who the intended users are (e.g., researchers, data providers, regulators)?...**
>
> Any copyright owners can deploy our watermarking when releasing their original datasets to a third party (e.g., AI training platforms, academic institutions and copyright certification systems). We have already talked about the real-world application of Glaze and NightShade in our introduction. To make the threat model more concrete, we will provide a detailed deployment scenario below.
>
> A company (called Alice) that collects a large proprietary dataset for autonomous driving research (e.g., dash cam video frames). She wants to open source a part of her dataset to promote innovation for community (e.g., Non-profit research organization), but also want to prevent unlicensed users to train a machine learning model on it successfully to protect her intellectual property.
>
> To achieve the above goals, Alice runs our poisoning+watermarking algorithm on every instance of their dataset, publishing the perturbed (i.e., protected) dataset which is unlearnable by standard models and obtains a secret, key dependent watermark signal. She publishes this on her GitHub under a permissive license, accompanied by a SHA256 hash so any recipient can verify integrity.
>
> A research lab (called Bob) registers on Alice’s portal and agree to a standard agreement for legal use of the dataset. After approval by Alice, Bob receives a secret key (e.g., a 128 bit seed) provided via Alice’s portal’s secure HTTPS channel. Furthermore, Bob also gains a pipeline (e.g., Python pre processing package) from Alice such that he can run the watermark detection to verify his identity and ensure that no file corruption. After the verification, Bob can run an algorithm designed by Alice (e.g., directly adding inverse unlearnable noise for each data) to remove the unlearnable poisons. If the pipeline receives wrong key or tampered file, the detection fails and the poisons cannot be removed to ensure the unlearnability.
>
> For a malicious user (called Chad), first, Chad can download the same public poisoned and watermarked dataset, but cannot train a good model on it because the dataset is unlearnable. If Chad tries to remove or tamper watermarks and unlearnable poisons without knowing the secret key, detection will fail.
>
> For key management, Alice can rotate keys per month and publish on her portal only to approved accounts (i.e., trusted users). Alice can also add a HMAC scheme to prevent potential forgery risks, which are raised by Reviewer RfGy, please see our rebuttal to Reviewer RfGy for details.
>
> Compared with licensing+hash approach (proposed by Review RfGy), our poisoning+watermarking approach can deter unauthorized use of the data in model training, giving data owners capability for prevention without inducing heavy overhead. Licensing+hash can only prove file integrity, fail to prevent a malicious user Chad to train a well-performed model when the data is leaked.
>
> **2. The writing and presentation could be improved for clarity and accessibility…**
>
> Thanks for your suggestion. $d$ and $\epsilon$ denotes the data dimension and the perturbation budget respectively, we will add the description in our Abstract and Introduction in the next version. For mathematical symbols, notations or definitions, according to the suggestion of Reviewer 9bm6, we will add a dedicated symbol table in the next version of our paper to enhance the readability. For experimental setup, we will add a detailed description in the Appendix to make it clearer.
>
> **3. The scope of the experimental evaluation is too limited…**
>
> Beyond CIFAR-10, we have already provided additional experiments on CIFAR-100 and TinyImageNet in Appendix C.1. To further validate the effectiveness of our method, for text dataset, we implement watermarking ($\epsilon_w=16/255$) in a backdoor attack on SST-2 dataset with BERT-base model [a], observing similar trends (Section 6) for this NLP task.
>
> ||Post-Poisoning|Poisoning-Concurrent|
> |-|-|-|
> |Length|ACC/ASR/AUROC| ACC/ASR/AUROC|
> |0|89.7/98.0/-|89.7/98.0/-|
> |100|89.8/97.8/0.697|89.6/97.2/0.969|
> |200|89.2/97.3/0.852|89.9/96.1/0.983|
> |400|89.6/96.2/0.931|89.3/90.5/0.998|
> |600|89.3/96.7/0.983|89.5/72.3/0.999|
>
> Moreover, according to the suggestion of Reviewer 91mv, we also conduct a preliminary experiment on WikiArt Dataset [b], a historical artists dataset to test our watermarking under Glaze protection [c]. Following the setting of Glaze, we finetune Stable Diffusion 2.1 to mimic victim artists’ style under Glaze protection with our post-poisoning and poisoning-concurrent watermarking respectively. We standardize the image of WikiArt to 768*768 and set watermarking dimension $q=5000$ with perturbation budget $\epsilon=8/255$. For evaluation, beyond AUROC, we follow [c] to test the protection performance using CLIP-based genre shift. Results show that our watermarking achieves high detection performance while keeping the protection of Glaze.
>
> ||AUROC|CLIP-based genre shift|
> |-|-|-|
> |Baseline|-|96.0\%|
> |Post-Poisoning|0.9999|95.8\%|
> |Poisoning-Concurrent|1.0000|95.3\%|
>
> Furthermore, for multi-modal data, following the experiment of [d], we use the MS-COCO dataset to generate text-image pairs, and conduct invisible backdoor attack in the image modality. We test both post-poisoning and poisoning-concurrent watermarking with $q=10000$ and $\epsilon=2/255$. Results demonstrate perfect detection performance (higher AUROC), decent backdoor performance (higher Benign Accuracy, higher ASR), and good image quality (higher PSNR, higher SSIM, lower MSE)
>
> ||AUROC|Benign Accuracy|ASR|PSNR|SSIM|MSE|
> |-|-|-|-|-|-|-|
> |Baseline|-|87.98|71.62|40.50|0.980|0.18|
> |Post-Poisoning|1.0000|88.02|71.93|36.78|0.974|0.25|
> |Poisoning-Concurrent|1.0000|87.95|68.23|39.41|0.978|0.23|
>
> Beyond these, according to the suggestion of Reviewer 9bm6, we further test our post-poisoning and poisoning-concurrent watermarking on ViT-B model [e] under UE, AP availability attacks and Narcissus, AdvSc backdoor attacks with different watermarking length $q$. The patch size is change from 16 to 4 in order to meet with the size of CIFAR-10. Similar conclusions can be summarized from the following results compared with discussions in Tables 1 and 2 in Section 6.2 on ResNet-18. Therefore, we believe our watermarking work well on modern architectures like Transformer.
>
> |Length|UE||AP||
> |-|-|-|-|-|
> |Acc/AUROC|Post-Poisoning|Poisoning-Concurrent|Post-Poisoning|Poisoning-Concurrent|
> |0 (Baseline)|13.82/-|13.82/-|10.99/-|10.99/-|
> |500|12.96/0.816|14.13/0.992|10.83/0.868|11.12/0.954|
> |1000|13.97/0.973|14.80/0.999|10.51/0.979|12.75/0.997|
> |1500|14.25/0.992|14.78/1.000|11.36/0.996|15.86/1.000|
> |2000|14.14/1.000|16.22/1.000|11.38/1.000|18.85/1.000|
> |2500|14.75/1.000|42.38/1.000|11.92/1.000|32.10/1.000|
> |3000|14.66/1.000|92.32/1.000|12.91/1.000|93.25/1.000|
>
> |Length|Narcissus||AdvSc||
> |-|-|-|-|-|
> |Acc/ASR/AUROC|Post-Poisoning|Poisoning-Concurrent|Post-Poisoning|Poisoning-Concurrent|
> |0 (Baseline)|95.13/94.32/-|95.13/94.32/-|94.02/93.58/-|94.02/93.58/-|
> |500|95.16/94.20/0.958|94.98/94.61/0.998|94.13/93.86/0.925|93.76/92.80/0.999|
> |1000|94.87/94.25/0.999|95.25/93.37/1.000|94.32/93.26/0.992|94.23/91.45/1.000|
> |1500|94.99/93.76/1.000|95.36/91.90/1.000|93.25/94.00/0.999|93.41/82.75/1.000|
> |2000|95.01/94.21/1.000|94.76/88.35/1.000|94.13/92.29/1.000|94.36/41.79/1.000|
> |2500|95.32/94.33/1.000|94.98/32.01/1.000|94.02/85.36/1.000|93.98/16.16/1.000|
> |3000|95.12/92.25/1.000|95.12/10.09/1.000|93.51/85.82/1.000|94.12/11.35/1.000|
>
>
> **4. data preprocessing, augmentation, and noise, none of which are accounted for in the current experiments.**
>
> We have considered many types of data augmentations, including Random Flip, Cutout, Color Jitter and Grayscale, as well as some regeneration attacks including VAE-based attack and GAN-based attack in Appendix D. Results in Table 10 show quite high robustness of our proposed watermarking. Furthermore, we also evaluate differential privacy noises with both Gaussian and Laplacian mechanism in Table 11, the poisoning utilities have been completely destroyed. Therefore, DP-based defenses are not applicable in our context. For data preprocessing like Normalization, our results (both theoretical and empirical) can be directly expanded with just some affine transformation. For other types of preprocessing, as our watermarking creator can be also served as poisoner (especially in poisoning-concurrent watermarking), they can just add watermarking after preprocessing to ensure the theoretical guarantee of our proposed watermarking.
>
> Ref:
>
> [a] Gan et al. Triggerless Backdoor Attack for NLP Tasks with Clean Labels. NAACL 2022.
>
> [b] Saleh and Elgammal. Large-scale classification of fine-art paintings: Learning the right metric on the right feature. arXiv:1505.00855.
>
> [c] Shan et al. Glaze: Protecting Artists from Style Mimicry by Text-to-Image Models. USENIX Security 23.
>
> [d] Zhang et al. BadCM: Invisible Backdoor Attack Against Cross-Modal Learning. IEEE TIP 2024.
>
> [e] Dosovitskiy et al. An Image is Worth 16x16 Words: Transformers for Image Recognition at Scale. ICLR 2021.

---

> > ### Comment · Reviewer_aXke · 2025-08-05
> >
> > I thank the authors for the rebuttal, which addressed most of my concerns. To improve clarity, I suggest including a concrete use case at the beginning of the paper to help readers better understand the motivation and application. I will increase my score accordingly.

---

> > > ### Author Response · Authors · 2025-08-06
> > >
> > > Thank you for raising your score and the time you have dedicated to reviewing our work. We will revise our manuscript based on your suggestion and add a Threat Model section including the concrete scenario of our watermarking under data poisoning.

---

### Official Review · Reviewer_9Mfj · 2025-06-30

**Clarity:** 3
**Significance:** 3
**Originality:** 4
**Rating:** 5
**Confidence:** 4

**Summary:**

1.  Theoretical soundness: Formal bounds on watermarking length for detectability and utility.

2.  Practical applicability: Demonstrated effectiveness on real-world poisoning attacks.

3.  Transparency: Addresses ethical concerns by enabling poisoned dataset identification.

**Questions:**

1. Generalizability: Could the framework extend to noisy-label or targeted poisoning attacks? If not, what are the fundamental barriers?

2. Scalability: How does watermarking perform on low-dimensional data (e.g., tabular) where  d is small?

3. Defense Mitigation: Could adaptive watermarking (e.g., dynamic watermark)   improve robustness against purification attacks?

**Ethical Concerns:**

["NO or VERY MINOR ethics concerns only"]

**Final Justification:**

The authors provide multiple perspectives on their work in their rebuttal, and I maintain my score.

**Limitations:**

Yes, limitations are acknowledged (e.g., necessity of large N, sensitivity to defenses).

**Quality:**

3

**Strengths And Weaknesses:**

Strengths

Quality: Rigorous theoretical analysis with proofs in appendices. Experiments validate claims across diverse attacks and models.

Clarity: Well-structured; theoretical and empirical sections are clearly separated. Figures/tables aid understanding.

Significance: Addresses a critical gap in responsible data poisoning (e.g., ownership verification, misuse prevention).

Originality: First work to integrate watermarking with poisoning attacks, with novel distinctions between post-poisoning and concurrent schemes.

Weaknesses

Evaluation Scope: Limited to imperceptible clean-label attacks; broader attack types (e.g., noisy-label) could strengthen generality.

Practical Constraints: Assumes large N and d for theoretical guarantees; impact on small-scale datasets is unclear.

Defense Robustness: Watermarking detectability degrades under diffusion purification (Table 12), though poisoning utility is also neutralized.

---

> ### Author Rebuttal · Authors · 2025-07-31
>
> Thanks for your positive feedback and valuable comments. We provide responses below to address your concerns.
>
> **1. Limited to imperceptible clean-label attacks; Could the framework extend to noisy-label or targeted poisoning attacks?**
>
> Thanks for your advice. Beyond clean-label attacks, we extend our experiments to dirty-label backdoor attacks in order to strengthen our findings. We evaluate three dirty-label attacks, including BadNet attack [a], Trojaning attack [b] and Invisible attack [c] on CIFAR-10 dataset. Following the original setting, in BadNet attack, we set the poisoning rate be 0.1, in Trojaning attack and Invisible attack, we set the poisoning rate be 0.05 and poisoning budget be $8/255$ under $L_{\inf}$ norm. We test both post-poisoning and poisoning-concurrent watermarking with length $q=1000$. Results shown that our watermarking algorithms embed well on these dirty-label backdoor attacks, achieve high test performance (AUROC) without affecting the poisoning utility (Acc and ASR).
>
> |Acc/ASR/AUROC|BadNet attack|Trojaning attack|Invisible attack|
> |-|-|-|-|
> |Baseline|0.52/0.93/-|0.91/0.96-|0.91/0.99/-|
> |Post-Poisoning|0.52/0.92/0.9986|0.92/0.96/0.9983|0.90/0.99/0.9989|
> |Poisoning-Concurrent|0.49/0.88/0.9998|0.89/0.97/0.9999|0.90/0.98/1.0000|
>
> For targeted poisoning attacks, we further conduct our experiments to two well-known attacks, Poison Frogs [d] and Witches Brew [e] on CIFAR-10 dataset with post-poisoning and poisoning-concurrent watermarking under $q=1000$. Similar trends shown in our results demonstrate the effectiveness of our watermarking under targeted poisoning attacks.
>
> |Target Acc/AUROC|Poison Frogs|Witches Brew|
> |-|-|-|
> |Baseline|0.53/-|0.82/-|
> |Post-Poisoning|0.53/0.9991|0.81/0.9990|
> |Poisoning-Concurrent|0.51/1.0000|0.79/1.0000|
>
> **2. Assumes large N and d for theoretical guarantees. How does watermarking perform on low-dimensional data (e.g., tabular) where d is small?**
>
> Although in theoretical results, it may require larger sample number $N$ and data dimension $d$, in practice, we find that our watermarking schemes work well for relatively low-dimension dataset. For instance, we test our watermarking under BadNet backdoor attacks on MNIST dataset ($d=768$) and USPS dataset ($d=256$). We set $q=400$ for MNIST and $q=150$ for USPS. Evaluation provided below demonstrate that, even for relatively low-dimensional dataset, both post-poisoning and poisoning-concurrent watermarking still display good detection performance as well as keeping backdoor utilities.
>
> |Acc/ASR/AUROC|MNIST|USPS|
> |-|-|-|
> |Baseline|0.98/1.00/-|0.62/0.87/-|
> |Post-Poisoning|0.98/1.00/0.9932|0.63/0.84/0.9815|
> |Poisoning-Concurrent|0.98/1.00/0.9977|0.61/0.78/0.9952|
>
> For tabular data, since most of them contain a large number of discrete/categorical variables (e.g., gender, regional code, risk level, etc.), the setting of imperceptible poisoning noises on such features often destroys semanticity. Therefore, it may not exist the scenario that using small perturbed data for copyright protection under tabular data. We may leave the watermarking of tabular data poisoning as the future work.
>
> **3. Defense Mitigation: Could adaptive watermarking (e.g., dynamic watermark) improve robustness against purification attacks?**
>
> Thanks for your valuable suggestion. For potential adaptive watermarking, we may train a surrogate network (encoder) to transfer universal watermarking into sample-wise version while keep the detection key be single. In this case, watermarks can be more elaborate and robust against stronger removal attacks, e.g., purification attacks. However, as our main focus on this paper is to provide a provable watermarking for data poisoning attacks, similar theoretical guarantees under more elaborate and adaptive watermarking may require more detailed analyses and rigorous proofs. We will add these discussions into the limitation and leave them as the future work.
>
> Ref:
>
> [a] Gu et al. BadNets: Identifying Vulnerabilities in the Machine Learning Model Supply Chain. arXiv:1708.06733.
>
> [b] Liu et al. Trojaning Attack on Neural Networks. NDSS 2018.
>
> [c] Li et al. Invisible Backdoor Attacks on Deep Neural Networks via Steganography and Regularization. arXiv:1909.02742.
>
> [d] Shafahi et al. Poison Frogs! Targeted Clean-Label Poisoning Attacks on Neural Networks.
> NeurIPS 2018.
>
> [e] Geiping et al. Witches’ Brew: Industrial Scale Data Poisoning via Gradient Matching. ICLR 2021.

---

> > ### Comment · Reviewer_9Mfj · 2025-08-03
> > **Thanks**
> >
> > Thanks for your reply, I learned a lot from it

---

> > > ### Author Response · Authors · 2025-08-06
> > >
> > > We thank for your reply and the time you have dedicated to reviewing our work. We will revise our manuscript based on your suggestions.

---

### Official Review · Reviewer_RfGy · 2025-06-30

**Clarity:** 3
**Significance:** 2
**Originality:** 2
**Rating:** 4
**Confidence:** 3

**Summary:**

This paper proposes a provable watermarking framework for poisoned datasets, addressing the question of how to embed a detectable signal alongside backdoor or availability poisons. The core idea is to add an additional perturbation to the poisoned data so that a trusted user with the correct key can verify whether the data was poisoned intentionally. The authors formalize the detection condition, provide theoretical bounds showing that the watermark is statistically detectable under budget constraints, and run extensive experiments across different poisoning methods (e.g., backdoors, unlearnable examples) on standard image benchmarks. They show that the watermark achieves high AUROC for detection while preserving the intended poisoning effect.

**Questions:**

The points raised in the **Weaknesses** section already outline my main questions and requests for clarification. In particular, I would appreciate clear answers or additional evidence regarding the real-world threat model and use cases, the practical management and security of the watermark key, the justification for separating poison and watermark instead of unifying them, and whether the method can withstand adaptive removal targeting only the watermark signal.

Addressing these questions would directly strengthen the practical relevance of the paper and would contribute to revising my rating positively.

**Ethical Concerns:**

["NO or VERY MINOR ethics concerns only"]

**Final Justification:**

The paper is theoretically sound, providing bounds for poison and watermarking perturbations, which I find timely and relevant. The authors have also addressed my concerns regarding adaptive attacks

**Limitations:**

Yes

**Quality:**

3

**Strengths And Weaknesses:**

### Strengths

- **Clear formal framework:** The paper presents a clean mathematical formulation, deriving explicit scaling laws for detection probability and showing how the watermark perturbation budget relates to detectability.

- **Sound empirical validation:** The experiments cover multiple threat types (backdoors, availability poisons) and confirm the theoretical soundness. They check that adding the watermark does not significantly degrade the poison’s intended effect.

- **Robustness checks:** The authors evaluate the watermark under basic data augmentations and simple regeneration defenses, and show that random incorrect keys produce only random-guess detection performance, which supports the claim that the key is non-trivial to guess.


Overall, the paper’s theoretical contributions are sound and well-supported by experiments. However, the idea of combining poisoning with a provable watermark is questionable.




### Weaknesses

1- **Practical Threat Model and Real-World Use**

The formal analysis and proofs are clear, but the practical story here still feels pretty abstract. The authors assume there’s a scenario where someone deliberately poisons their own data to protect it (like using unlearnable examples) and wants to help “good” users avoid accidentally using that poison by adding a watermark and secret key. But in practice, it’s not obvious who would really do this. If you want to protect your data from misuse, you can just license it properly, publish a hash, or provide a clean version for trusted partners — you don’t necessarily need an extra hidden signal inside the data.

I’d like to see a concrete example of how this actually plays out: Who exactly is sharing poisoned data and a watermark key? Who runs the detection? How is the key managed and trusted in real-world distribution? For instance — an artist uploading watermarked UE images to the internet doesn’t really have a clear way to send keys to every possible good-faith user. Without this piece, it’s hard to see how this idea moves beyond an interesting theoretical guarantee.

Overall, I’d encourage the authors to ground this threat model in a realistic scenario and explain who the real users are and how they’d benefit. This would help connect the theory to a clearer real-world story.

---

2- **Use of Separate Watermark and Poison Perturbations**

The authors do not sufficiently motivate why the poison itself cannot be designed to serve as a watermark, or vice versa. For backdoor attacks, it is well established that the backdoor trigger itself naturally functions as a watermark: for example, [arXiv:1802.04633v3], [arXiv:2208.00563v2], and [arXiv:2305.12502] all demonstrate that hidden backdoor patterns can double as watermarks to prove intellectual property ownership without requiring any additional watermark noise.

More generally, for both backdoor and availability attacks, the paper does not analyze whether the watermark perturbation could itself be crafted to serve as the poison — that is, whether a unified perturbation could achieve both detectability and attack effectiveness simultaneously. For availability poisoning, this might require designing the unlearnable noise to embed a structured, provable pattern, rather than adding a separate signal. Such a joint design could reduce the total perturbation budget, simplify implementation, and remove the need to split the signal across separate dimensions or budgets.

Overall, I recommend the authors clarify and empirically test whether separate perturbations are truly necessary for both attack classes, or whether a combined poison–watermark perturbation could achieve the same provable guarantees while simplifying the threat model and deployment.

---

3- **Key Management and Forgery Risk**

The proposed watermark relies entirely on the secrecy of the key vector \(\zeta\), which determines the detection mechanism. While the paper demonstrates that using a random incorrect key results in random-guess performance (AUROC ≈ 0.5), it does not discuss how this key should be securely generated, stored, or distributed to legitimate users in real-world scenarios. If the key were to leak or be exposed during dataset sharing or verification, an adversary could potentially remove the watermark by reversing or masking the known perturbation, or forge a similar watermark on unrelated clean data.
To strengthen this aspect, I suggest the authors clarify:
- How the key \(\zeta\) should be generated and kept confidential in practical deployment.
- How authorized users should receive or verify the key without risking exposure.
- Whether there are simple binding or signature mechanisms that could make the watermark non-forgeable if the key is ever leaked.

Without addressing this, the scheme’s detectability guarantee depends entirely on perfect secrecy of the key, which may be unrealistic in open data sharing settings.

---

4- **Adaptive Removal Risk Due to Poison–Watermark Separation**

A core feature of the proposed method is that the poisoning and watermarking perturbations are explicitly separated into different sets of dimensions (or partially separated). While this makes the theoretical analysis clear, it also creates a practical risk: an adaptive attacker who knows or estimates which dimensions carry the watermark could remove only the watermark perturbation while preserving the poison effect. For example, an attacker could analyze multiple samples to identify the watermark dimensions, then mask, smooth, or overwrite only those dimensions, leaving the poison untouched. If this works, the watermark detectability would fail while the attack’s intended effect would remain intact, undermining the claimed provable ownership.

Recent work ([arXiv:2309.16952v2]) shows that even stronger adaptive attacks are feasible: attackers can use optimization-based surrogate key recovery and iterative removal to erase hidden watermarks with minimal visible distortion. This result demonstrates that modern watermark removal is not purely theoretical but practical, and particularly relevant for sparse, separable perturbations like the one proposed here.
To clarify real-world robustness, I strongly recommend that the authors add an experiment that directly tests this threat. Concretely, they could: (1) simulate an attacker who removes or masks only the known watermark dimensions; (2) test whether the watermark detection AUROC drops to random; (3) verify whether the poison’s effect (e.g., attack success rate or degradation of model performance) stays intact; and (4) optionally compare with a simple optimization-based removal approach as demonstrated by Lukas et al. If the watermark is easily removed while the poison remains, this would show that the explicit separation makes the signal vulnerable and highlight the need to consider partial overlap or a hybrid joint design.

---

> ### Author Rebuttal · Authors · 2025-07-31
>
> Thanks for your insightful comments and valuable suggestions. We provide responses below to address your concerns.
>
> **1. Practical Threat Model and Real-World Use.**
>
> Any copyright owners can deploy our watermarking when releasing their original datasets to a third party (e.g., AI training platforms, academic institutions and copyright certification systems). To make the threat model more concrete, we will provide a detailed deployment scenario below.
>
> A company (called Alice) that collects a large proprietary dataset for autonomous driving research (e.g., dash cam video frames). She wants to open source a part of her dataset to promote innovation for community (e.g., Non-profit research organization), but also want to prevent unlicensed users to train a machine learning model on it successfully to protect her intellectual property.
>
> To achieve the above goals, Alice runs our poisoning+watermarking algorithm on every instance of their dataset, publishing the perturbed (i.e., protected) dataset which is unlearnable by standard models and obtains a secret, key dependent watermark signal. She publishes this on her GitHub under a permissive license, accompanied by a SHA256 hash so any recipient can verify integrity.
>
> A research lab (called Bob) registers on Alice’s portal and agree to a standard agreement for legal use of the dataset. After approval by Alice, Bob receives a secret key (e.g., a 128 bit seed) provided via Alice’s portal’s secure HTTPS channel. Furthermore, Bob also gains a pipeline (e.g., Python pre processing package) from Alice such that he can run the watermark detection to verify his identity and ensure that no file corruption. After the verification, Bob can run an algorithm designed by Alice (e.g., directly adding inverse unlearnable noise for each data) to remove the unlearnable poisons. If the pipeline receives wrong key or tampered file, the detection fails and the poisons cannot be removed to ensure the unlearnability.
>
> For a malicious user (called Chad), first, Chad can download the same public poisoned and watermarked dataset, but cannot train a good model on it because the dataset is unlearnable. If Chad tries to remove or tamper watermarks and unlearnable poisons without knowing the secret key, detection will fail.
>
> For key management, Alice can rotate keys per month and publish on her portal only to approved accounts (i.e., trusted users). Alice can also add a HMAC scheme to prevent potential forgery risks (please see our response to Q3 for the detailed answer).
>
> Compared with licensing and hash approach as you suggested, our poisoning and watermarking approach can deter unauthorized use of the data in model training, giving data owners capability for prevention without inducing heavy overhead. Licensing and hash can only prove file integrity, fail to prevent a malicious user Chad to train a well-performed model when the data is leaked.
>
> **2. Use of Separate Watermark and Poison Perturbations.**
>
> In practice, the poisons (both backdoor attacks and availability attacks) are sample-wise noises, that means every sample is injected with a specific poisoning noise. However, practical watermarking proposed in our scheme becomes universal, which makes it easier for key management, using a single detection key. Therefore, if we try to use the combined poison–watermark perturbation, we also need to make poisons become universal noises. Unfortunately, if we want to design universal poisons for the dataset, the poisoning utility is completely destroyed. As the result demonstrated, Universal UE and AP will recover the test accuracy to almost 90\%, completely violate the goal of availability attacks: make dataset unlearnable. Similar trends also have been shown for universal backdoor attacks, the ASR drop to about 10\%, resulting in the failure of backdoor triggers.
>
> |Acc|UE|AP|
> |-|-|-|
> |Sample-wise(baseline)|10.79|8.53|
> |Universal|88.75|89.34|
>
> |Acc/ASR|Narcissus|AdcSc|
> |-|-|-|
> |Sample-wise(baseline)|94.69/95.04|92.80/95.53|
> |Universal|95.13/11.02|93.89/10.00|
>
> For potential direct watermarking with backdoor attacks ([arXiv:1802.04633v3], [arXiv:2208.00563v2], and [arXiv:2305.12502]), we highlight that these works are model watermarking, which embed watermarking signals into model parameters to achieve model ownership verification. However, our paper considers the provable watermarking under poisoned dataset (rather than model), to discern legitimate users and avoid unauthorized use for potential malicious users. Even if model parameters are unavailable, a legitimate user can still detect watermarking of the dataset with their secret key.
>
> **3. Key Management and Forgery Risk.**
>
> As our answer in Q1, Alice can rotate keys per month and publish on their portal only to approved accounts (i.e., trusted users). Alice can also add a HMAC scheme to prevent potential forgery risks. Specifically, we can separate keys into generation key $k_{gen}$ and authentication key $k_{auth}$, where $k_{gen}$ is completely same with our paper and correlate with injected watermarks $w_i$ for every data $x_i$. For each perturbed $ \hat{x_i}$ with $w_i$, we can compute an additional tag $t_i$ by HMAC under $k_{auth}$, i.e., $t_i = HMAC_{k_{auth}}(id_i, \hat{x_i})$, where $id_i$ is a unique identifier for the image $x_i$ (e.g., index). After that, we store the ($id_i, t_i$) pair (e.g., through a sidecar JSON) for latter detection. In watermarking detection, beyond traditional detection using $k_{gen}$ and $ \hat{x_i}$, we also verify the tag with $t_i$ and $t_i = HMAC_{k_{auth}}(id_i, \hat{x_i})$ to avoid the potential forgery attacks. In this case, even if the generation key $k_{gen}$ leaks, an attacker cannot forge a new valid $(x_i, t_i)$ pair as they lack the authentication key $k_{auth}$. We can keep $k_{auth}$ in a secure enclave and rotate it independently with $k_{gen}$ to enhance the security.
>
> **4. Adaptive Removal Risk Due to Poison–Watermark Separation.**
>
> Thanks for your suggestion. We conduct additional experiments on UE and AP with direct masking of the known watermarking dimensions (Masking), as well as the adversarial noising proposed by [arXiv:2309.16952v2]. We test both post-poisoning and poisoning-concurrent watermarking under $q=2000$. As the results shown below, although the detection performance (AUROC) drops, the utility of UE and AP also degrade significantly. The underlying reasons may be that availability attacks are designed with potential linear shortcut features [a,b], the masking of watermarking dimensions somehow destroy these linear features, undermining the unlearnability (low Acc). Adversarial Noising further destroys the poisoning utility as availability attacks are theoretically removed by perfect adversarial training [c]. Therefore, these adaptive removal attacks fail to maintain the poisoning utility, making them not applicable in our cases, similar with DP noises and purifications discussed in our Appendix D.
>
> |Acc/AUROC| Baseline |Masking | Adversarial Noising |
> |-|-|-|-|
> |UE(Post-Poisoning)|9.06/0.9992|60.71/0.4998|72.90/0.5893|
> |AP(Post-Poisoning)| 10.48/0.9987|56.85/0.5005|76.21/0.5616|
> |UE(Poisoning-Concurrent)| 10.03/1.0000|55.49/0.5014|68.37/0.6206|
> |AP(Poisoning-Concurrent)| 38.62/1.0000|59.87/0.5002|74.63/0.5833|
>
>
> Ref:
>
> [a] Yu et al. Availability Attacks Create Shortcuts. KDD 2022.
>
> [b] Zhu et al. Defense of Unlearnable Examples. AAAI 2024.
>
> [c] Tao et al. Better Safe Than Sorry: Preventing Delusive Adversaries with Adversarial Training. NeurIPS 2021.

---

> > ### Comment · Reviewer_RfGy · 2025-08-01
> >
> > I thank the authors for their detailed rebuttal, which addresses my initial concerns. Taking into account the clarifications provided, the additional experiments, and the novelty of the proposed framework, I am increasing my rating.

---

> > > ### Author Response · Authors · 2025-08-06
> > >
> > > Thank you for raising your score and the time you have dedicated to reviewing our work. We will revise our manuscript based on your suggestions.

---

### Official Review · Reviewer_91mv · 2025-07-01

**Clarity:** 2
**Significance:** 3
**Originality:** 3
**Rating:** 4
**Confidence:** 3

**Summary:**

This paper proposes using provable watermarking to allow creators of harmlessly poisoned data to transparently declare their modifications. The authors introduce two schemes: post-poisoning and poisoning-concurrent watermarking. They provide theoretical guarantees for the detectability of the watermark and the preservation of the poison's utility, tying these properties to the watermark's length. The theories are then validated on several common clean-label data poisoning attacks.

**Questions:**

The lack of error bars is a key weakness. Can you provide results from multiple random seeds for even a small subset of experiments to demonstrate the variance of your metrics?

How does your theoretical framework apply to the complex perturbations of poisoning-for-good tools like Nightshade, which motivated your work?

The DP noise defense invalidates the model's utility. Can you discuss a more realistic adaptive attack where an adversary tries to remove the watermark while preserving model utility?

**Ethical Concerns:**

["NO or VERY MINOR ethics concerns only"]

**Final Justification:**

Thanks for the rebuttal.

**Limitations:**

Yes.

**Quality:**

2

**Strengths And Weaknesses:**

Strengths:
The paper addresses the important problem of transparency for harmlessly data poisoning. The application of provable watermarking is meaningful to this emerging area.

The work is theoretically sound, providing clear, provable bounds for the proposed schemes under both sample-wise and universal settings. The paper is clearly structured.

Weaknesses:
The paper is motivated by recent "data poisoning for good" methods like Nightshade and Glaze. However, the experiments are conducted on more traditional clean-label backdoor and availability attacks. It is not immediately clear if the paper's theoretical model, which assumes imperceptible additive noise, directly applies to the complex, style-based perturbations of tools like Glaze. Experiments of such an attack would have made the connection stronger.

The experiments are conducted on small and traditional architectures (e.g., ResNet-18, VGG-19). This limits the generalizability of the findings to the large-scale scenarios that motivate the work.

The experiments lack statistical rigor. Results are based on single runs without error bars, as noted in the appendix, which makes it difficult to assess the variance and significance of the empirical claims.

---

> ### Author Rebuttal · Authors · 2025-07-31
>
> Thanks for your positive feedback and valuable comments. We provide responses below to address your concerns.
>
> **1. How does your theoretical framework apply to the complex perturbations of poisoning-for-good tools like Glaze and Nightshade?**
>
> Thanks for your valuable suggestion. We conduct a preliminary experiment on WikiArt Dataset [a], a historical artists dataset to test our watermarking under Glaze protection [b]. Following the setting of Glaze, we finetune Stable Diffusion 2.1 to mimic victim artists’ style under Glaze protection with our post-poisoning and poisoning-concurrent watermarking respectively. We standardize the image of WikiArt to 768*768 and set watermarking dimension $q=5000$ with perturbation budget $\epsilon=8/255$. For evaluation, beyond AUROC, we follow [b] to test the protection performance using CLIP-based genre shift. Results show that our watermarking achieves high detection performance while keeping the protection of Glaze.
>
> ||AUROC|CLIP-based genre shift|
> |-|-|-|
> |Baseline|-|96.0\%|
> |Post-Poisoning|0.9999|95.8\%|
> |Poisoning-Concurrent|1.0000|95.3\%|
>
> **2. The experiments are conducted on small and traditional architectures (e.g., ResNet-18, VGG-19). This limits the generalizability of the findings to the large-scale scenarios that motivate the work.**
>
> We have evaluated many well-known architectures including ResNet, VGG, DenseNet, MobileNet and Wide-ResNet in Appendix C.2. We further test our post-poisoning and poisoning-concurrent watermarking on ViT-B model [c] under UE, AP availability attacks and Narcissus, AdvSc backdoor attacks with different watermarking length $q$. The patch size is change from 16 to 4 in order to meet with the size of CIFAR-10. Similar conclusions can be summarized from the following results compared with discussions in Tables 1 and 2 in Section 6.2 on ResNet-18. Therefore, we believe our watermarking work well on modern architectures like Transformer.
>
> |Length|UE||AP||
> |-|-|-|-|-|
> |Acc/AUROC|Post-Poisoning|Poisoning-Concurrent|Post-Poisoning|Poisoning-Concurrent|
> |0 (Baseline)|13.82/-|13.82/-|10.99/-|10.99/-|
> |500|12.96/0.816|14.13/0.992|10.83/0.868|11.12/0.954|
> |1000|13.97/0.973|14.80/0.999|10.51/0.979|12.75/0.997|
> |1500|14.25/0.992|14.78/1.000|11.36/0.996|15.86/1.000|
> |2000|14.14/1.000|16.22/1.000|11.38/1.000|18.85/1.000|
> |2500|14.75/1.000|42.38/1.000|11.92/1.000|32.10/1.000|
> |3000|14.66/1.000|92.32/1.000|12.91/1.000|93.25/1.000|
>
> |Length|Narcissus||AdvSc||
> |-|-|-|-|-|
> |Acc/ASR/AUROC|Post-Poisoning|Poisoning-Concurrent|Post-Poisoning|Poisoning-Concurrent|
> |0 (Baseline)|95.13/94.32/-|95.13/94.32/-|94.02/93.58/-|94.02/93.58/-|
> |500|95.16/94.20/0.958|94.98/94.61/0.998|94.13/93.86/0.925|93.76/92.80/0.999|
> |1000|94.87/94.25/0.999|95.25/93.37/1.000|94.32/93.26/0.992|94.23/91.45/1.000|
> |1500|94.99/93.76/1.000|95.36/91.90/1.000|93.25/94.00/0.999|93.41/82.75/1.000|
> |2000|95.01/94.21/1.000|94.76/88.35/1.000|94.13/92.29/1.000|94.36/41.79/1.000|
> |2500|95.32/94.33/1.000|94.98/32.01/1.000|94.02/85.36/1.000|93.98/16.16/1.000|
> |3000|95.12/92.25/1.000|95.12/10.09/1.000|93.51/85.82/1.000|94.12/11.35/1.000|
>
> Furthermore, for text dataset, we implement watermarking ($\epsilon_w=16/255$) in a backdoor attack on SST-2 dataset with BERT-base model [d], observing similar trends (Section 6) for this NLP task.
>
> ||Post-Poisoning|Poisoning-Concurrent|
> |-|-|-|
> |Length|ACC/ASR/AUROC| ACC/ASR/AUROC|
> |0|89.7/98.0/-|89.7/98.0/-|
> |100|89.8/97.8/0.697|89.6/97.2/0.969|
> |200|89.2/97.3/0.852|89.9/96.1/0.983|
> |400|89.6/96.2/0.931|89.3/90.5/0.998|
> |600|89.3/96.7/0.983|89.5/72.3/0.999|
>
> Moreover, according to the suggestion of Reviewer aXke, we further conduct experiments for multi-modal data. Following the experiment of [e], we use the MS-COCO dataset to generate text-image pair, and conduct invisible backdoor attack in the image modality. We test both post-poisoning and poisoning-concurrent watermarking with $q=10000$ and $\epsilon=2/255$. Results demonstrate perfect detection performance (higher AUROC), decent backdoor performance (higher Benign Accuracy, higher ASR), and good image quality (higher PSNR, higher SSIM, lower MSE).
>
> ||AUROC|Benign Accuracy|ASR|PSNR|SSIM|MSE|
> |-|-|-|-|-|-|-|
> |Baseline|-|87.98|71.62|40.50|0.980|0.18|
> |Post-Poisoning|1.0000|88.02|71.93|36.78|0.974|0.25|
> |Poisoning-Concurrent|1.0000|87.95|68.23|39.41|0.978|0.23|
>
> **3. The lack of error bars.**
>
> Thanks for your advice. We run both post-poisoning and poisoning-concurrent watermarking with $q=1000$ under UE, AP availability attacks and Narcissus, AdvSc backdoor attacks for five times on different random seeds. Results shown below demonstrate that our watermarking are not sensitive to random seeds.
>
> |Acc/ASR/AUROC|UE|AP|Narcissus|AdvSc|
> |-|-|-|-|-|
> |Post-Poisoning|11.42$\pm$0.72/-/0.9503$\pm$0.0065|10.56$\pm$0.31/-/0.9735$\pm$0.0112|94.43$\pm$0.23/92.36$\pm$1.05/0.9974$\pm$0.0008|92.94$\pm$0.32/94.93$\pm$1.90/0.9806$\pm$0.0057|
> |Poisoning-Concurrent|9.97$\pm$0.66/-/0.9993$\pm$0.0006|20.95$\pm$2.32/-/0.9950$\pm$0.0021|94.36$\pm$0.18/91.85$\pm$1.74/0.9990$\pm$0.0006|92.99$\pm$1.05/85.22$\pm$3.47/0.9995$\pm$0.0003|
>
> **4. The DP noise defense invalidates the model's utility. Can you discuss a more realistic adaptive attack where an adversary tries to remove the watermark while preserving model utility?**
>
> Our watermarking algorithms are either robust to existing attacks, or induce attacks to degrade poisoning utilities, which means our watermarking under poisoning are effective and without negative impact. For potential adaptive attacks, we can consider a scenario that requiring higher ability for attackers: Assuming that attackers have both watermarked and non-watermarked images, and they can detect them effectively (e.g., obtaining the secret key). Our watermarks can be removed by simply subtract the average of watermarked images and which of the non-watermarked images as our watermarking are universal typically. For more finely designed adaptive attacks, we will take them as the future work.
>
>
> Ref:
>
> [a] Saleh and Elgammal. Large-scale classification of fine-art paintings: Learning the right metric on the right feature. arXiv:1505.00855.
>
> [b] Shan et al. Glaze: Protecting Artists from Style Mimicry by Text-to-Image Models. USENIX Security 23.
>
> [c] Dosovitskiy et al. An Image is Worth 16x16 Words: Transformers for Image Recognition at Scale. ICLR 2021.
>
> [d] Gan et al. Triggerless Backdoor Attack for NLP Tasks with Clean Labels. NAACL 2022.
>
> [e] Zhang et al. BadCM: Invisible Backdoor Attack Against Cross-Modal Learning. IEEE TIP 2024.

---

### Official Review · Reviewer_9bm6 · 2025-07-01

**Clarity:** 3
**Significance:** 3
**Originality:** 3
**Rating:** 4
**Confidence:** 3

**Summary:**

The author explores the use of watermarking to enable poisoners to provably declare the presence of poisoning, thereby promoting transparency and reducing potential disputes. The paper introduces two provable and practical watermarking approaches for data poisoning.

**Questions:**

How the watermark performs on other model families, such as transformers or GNNs?

**Ethical Concerns:**

["NO or VERY MINOR ethics concerns only"]

**Final Justification:**

Thanks for the author’s clarification, which resolves my concern. Still, compared with other submissions, I will stick to my original evaluation and decision.

**Limitations:**

There are several limitations should be addressed to further strengthen the work:

1. The stated goal of watermarking in data poisoning is to allow authorized users to detect whether a dataset has been poisoned using a secret key. However, this threat model appears unrealistic. If the data provider is trustworthy, they would claim whether poisoning is present without requiring watermark-based detection. Conversely, if the provider is untrustworthy, there is little reason to expect them to share the key needed for detection, undermining the proposed verification mechanism.

2. The paper lacks discussion on how the proposed watermarking techniques generalize to different victim models, such as transformers, or GNNs. Evaluating a broader range of architectures would enhance the practical relevance of the work.

3. There is significant notation overload. Many symbols are introduced before being properly defined, which impairs readability. A dedicated symbol table or an overview figure early in the paper would help readers more effectively.

**Quality:**

3

**Strengths And Weaknesses:**

Strengths
1. The idea is novel.

Weaknesses
1. Unrealistic threat model.
2. No evaluation on other model types.
3. Dense, undefined notation—add a symbol table.

---

> ### Author Rebuttal · Authors · 2025-07-31
>
> Thanks for your positive feedback and valuable comments. We provide responses below to address your concerns.
>
> **1. Unrealistic threat model.**
>
> The poisoning considered in our paper are more refer to copyright protection, which has been discussed in the Introduction. Concretely, poisoner itself is a data provider. Watermarking does not consider whether the data provider is trusted, but rather whether users trustworthy. If they are trustworthy, watermarked dataset will inform users that the data is protected from poisoning, allowing them to reverse the operation to extract a clean dataset for normal use. For an malicious and untrusted user, he cannot obtain authorization as no detection key holds. Therefore, he can only get an unlearnable dataset. To make the threat model more concrete, we will provide a detailed deployment scenario below.
>
> A company (called Alice) that collects a large proprietary dataset for autonomous driving research (e.g., dash cam video frames). She wants to open source a part of her dataset to promote innovation for community (e.g., Non-profit research organization), but also want to prevent unlicensed users to train a machine learning model on it successfully to protect her intellectual property.
>
> To achieve the above goals, Alice runs our poisoning+watermarking algorithm on every instance of their dataset, publishing the perturbed (i.e., protected) dataset which is unlearnable by standard models and obtains a secret, key dependent watermark signal. She publishes this on her GitHub under a permissive license, accompanied by a SHA256 hash so any recipient can verify integrity.
>
> A research lab (called Bob) registers on Alice’s portal and agree to a standard agreement for legal use of the dataset. After approval by Alice, Bob receives a secret key (e.g., a 128 bit seed) provided via Alice’s portal’s secure HTTPS channel. Furthermore, Bob also gains a pipeline (e.g., Python pre processing package) from Alice such that he can run the watermark detection to verify his identity and ensure that no file corruption. After the verification, Bob can run an algorithm designed by Alice (e.g., directly adding inverse unlearnable noise for each data) to remove the unlearnable poisons. If the pipeline receives wrong key or tampered file, the detection fails and the poisons cannot be removed to ensure the unlearnability.
>
> For a malicious user (called Chad), first, Chad can download the same public poisoned and watermarked dataset, but cannot train a good model on it because the dataset is unlearnable. If Chad tries to remove or tamper watermarks and unlearnable poisons without knowing the secret key, detection will fail.
>
> For key management, Alice can rotate keys per month and publish on her portal only to approved accounts (i.e., trusted users). Alice can also add a HMAC scheme to prevent potential forgery risks, which are raised by Reviewer RfGy, please see our rebuttal to Reviewer RfGy for details.
>
> Compared with licensing+hash approach (proposed by Review RfGy), our poisoning+watermarking approach can deter unauthorized use of the data in model training, giving data owners capability for prevention without inducing heavy overhead. Licensing+hash can only prove file integrity, fail to prevent a malicious user Chad to train a well-performed model when the data is leaked.
>
> **2. No evaluation on other model types. How the watermark performs on other model families, such as transformers or GNNs?**
>
> We have evaluated many well-known architectures including ResNet, VGG, DenseNet, MobileNet and Wide-ResNet in Appendix C.2. We further test our post-poisoning and poisoning-concurrent watermarking on ViT-B model [a] under UE, AP availability attacks and Narcissus, AdvSc backdoor attacks with different watermarking length $q$. The patch size is change from 16 to 4 in order to meet with the size of CIFAR-10. Similar conclusions can be summarized from the following results compared with discussions in Tables 1 and 2 in Section 6.2 on ResNet-18. Therefore, we believe our watermarking work well on modern architectures like Transformer.
>
> |Length|UE||AP||
> |-|-|-|-|-|
> |Acc/AUROC|Post-Poisoning|Poisoning-Concurrent|Post-Poisoning|Poisoning-Concurrent|
> |0 (Baseline)|13.82/-|13.82/-|10.99/-|10.99/-|
> |500|12.96/0.816|14.13/0.992|10.83/0.868|11.12/0.954|
> |1000|13.97/0.973|14.80/0.999|10.51/0.979|12.75/0.997|
> |1500|14.25/0.992|14.78/1.000|11.36/0.996|15.86/1.000|
> |2000|14.14/1.000|16.22/1.000|11.38/1.000|18.85/1.000|
> |2500|14.75/1.000|42.38/1.000|11.92/1.000|32.10/1.000|
> |3000|14.66/1.000|92.32/1.000|12.91/1.000|93.25/1.000|
>
> |Length|Narcissus||AdvSc||
> |-|-|-|-|-|
> |Acc/ASR/AUROC|Post-Poisoning|Poisoning-Concurrent|Post-Poisoning|Poisoning-Concurrent|
> |0 (Baseline)|95.13/94.32/-|95.13/94.32/-|94.02/93.58/-|94.02/93.58/-|
> |500|95.16/94.20/0.958|94.98/94.61/0.998|94.13/93.86/0.925|93.76/92.80/0.999|
> |1000|94.87/94.25/0.999|95.25/93.37/1.000|94.32/93.26/0.992|94.23/91.45/1.000|
> |1500|94.99/93.76/1.000|95.36/91.90/1.000|93.25/94.00/0.999|93.41/82.75/1.000|
> |2000|95.01/94.21/1.000|94.76/88.35/1.000|94.13/92.29/1.000|94.36/41.79/1.000|
> |2500|95.32/94.33/1.000|94.98/32.01/1.000|94.02/85.36/1.000|93.98/16.16/1.000|
> |3000|95.12/92.25/1.000|95.12/10.09/1.000|93.51/85.82/1.000|94.12/11.35/1.000|
>
> Furthermore, according to the suggestion of Reviewer 91mv, we conduct a preliminary experiment on WikiArt Dataset [b], a historical artists dataset to test our watermarking under Glaze protection [c]. Following the setting of Glaze, we finetune Stable Diffusion 2.1 to mimic victim artists’ style under Glaze protection with our post-poisoning and poisoning-concurrent watermarking respectively. We standardize the image of WikiArt to 768*768 and set watermarking dimension $q=5000$ with perturbation budget $\epsilon=8/255$. For evaluation, beyond AUROC, we follow [c] to test the protection performance using CLIP-based genre shift. Results show that our watermarking achieves high detection performance while keeping the protection of Glaze.
>
> ||AUROC|CLIP-based genre shift|
> |-|-|-|
> |Baseline|-|96.0\%|
> |Post-Poisoning|0.9999|95.8\%|
> |Poisoning-Concurrent|1.0000|95.3\%|
>
> Moreover, according to the suggestion of Reviewer aXke, we further conduct experiments for multi-modal data. Following the experiment of [d], we use the MS-COCO dataset to generate text-image pairs, and conduct invisible backdoor attack in the image modality. We test both post-poisoning and poisoning-concurrent watermarking with $q=10000$ and $\epsilon=2/255$. Results demonstrate perfect detection performance (higher AUROC), decent backdoor performance (higher Benign Accuracy, higher ASR), and good image quality (higher PSNR, higher SSIM, lower MSE).
>
> ||AUROC|Benign Accuracy|ASR|PSNR|SSIM|MSE|
> |-|-|-|-|-|-|-|
> |Baseline|-|87.98|71.62|40.50|0.980|0.18|
> |Post-Poisoning|1.0000|88.02|71.93|36.78|0.974|0.25|
> |Poisoning-Concurrent|1.0000|87.95|68.23|39.41|0.978|0.23|
>
> **3. There is significant notation overload. Many symbols are introduced before being properly defined, which impairs readability. A dedicated symbol table or an overview figure early in the paper would help readers more effectively.**
>
> Thanks for your valuable suggestion. We will add a dedicated symbol table in the next version of our paper to enhance the readability.
>
> Ref:
>
> [a] Dosovitskiy et al. An Image is Worth 16x16 Words: Transformers for Image Recognition at Scale. ICLR 2021.
>
> [b] Saleh and Elgammal. Large-scale classification of fine-art paintings: Learning the right metric on the right feature. arXiv:1505.00855.
>
> [c] Shan et al. Glaze: Protecting Artists from Style Mimicry by Text-to-Image Models. USENIX Security 23.
>
> [d] Zhang et al. BadCM: Invisible Backdoor Attack Against Cross-Modal Learning. IEEE TIP 2024.

---

### Decision · Program_Chairs · 2025-09-17

**Decision:**

Accept (poster)

**Comment:**

This paper proposes a provable watermarking approach enabling creators of harmlessly poisoned data to transparently declare their modifications. Before the rebuttal, reviewers recognized the timeliness and practical significance of the problem, as well as the soundness of the theoretical analyses. Nonetheless, they raised concerns about the limited evaluation against stronger or more adaptive attacks and the unclear generalizability of the method. During the rebuttal and discussion periods, the authors addressed these points through additional experiments and extended explanations. Balancing the paper’s clear strengths in problem significance and theoretical rigor against its remaining limitations, I recommend acceptance.